

# Microphysics of radiation fog and estimation of fog deposition velocity for atmospheric dispersion applications

Rachid Abida[1], Narendra Nelli[1], Diana Francis[1*], Olivier Masson[2], Ricardo Fonseca[1], Emmanuel Bosc[3] and Marouane Temimi [4]

[1] Environmental and Geophysical Sciences (ENGEOS) Lab, Khalifa University of Science and Technology, P. O. Box 127788, Abu Dhabi, United Arab Emirates.

[2] Institut de Radioprotection et de Sûreté Nucléaire (IRSN), LEREN, Cadarache, 13115 Saint Paul-Lez-Durance, France

[3] Federal Authority for Nuclear Regulation (FANR), P.O. Box: 112021, Abu Dhabi, United Arab Emirates.

[4] Department of Civil, Environmental and Ocean Engineering (CEOE), Stevens Institute of Technology, Hoboken, NJ 07030, United States of America

*Correspondence to*: Diana Francis (diana.francis@ku.ac.ae)

## Abstract

Fog augments the wet deposition of airborne particles entrained in its hydrometeors. This article aims to characterize fog deposition processes around the Barakah nuclear power plant (BNPP), in the United Arab Emirates (UAE), and assess the potential impact of fog on the deposition rate of radionuclides in case of an accidental release. To this end, the microphysics of twelve radiation fog events, typical in such arid climate, were measured during the winter seasons of 2021 and 2022 using a fog monitor that was deployed at the BNPP. The impact of fog deposition on the settling of radionuclides is investigated based on model simulations using the Weather Research Forecasting (WRF) model with the MYJ PBL scheme and FLEXPART.

All fog events are found to share a common feature of a bimodal distribution in droplet number concentration ($N_c$), with modes at 4.5 $\mu m$ and 23.16 $\mu m$. It was pointed out that despite the high proportion of smaller droplets in the fog associated with the fine mode, the greatest contribution to the liquid water content (LWC) comes essentially from medium to large droplets between 10 and 35 $\mu m$. The deposition flux of fog water at the site and the fog droplet deposition velocity were estimated using an Eddy Covariance (EC) onsite. Typical mean values for fog droplet deposition velocity are found to range between 2.11 and 7.87 cm s$^{-1}$. The modeling results show that fog deposition contributed by 30-40% to the total ground deposition of $^{137}$Cs, highlighting the



importance of incorporating fog deposition as an additional scavenging mechanism in dispersion
modeling under foggy conditions.
**Keywords:** Radiation fog, fog microphysics, desert regions, fog deposition velocity, fog water
deposition flux, fog monitor, Weather Research Forecasting (WRF) model.
**1.  Introduction**
Following the Fukushima accident and its consequences in Japan, it was recognized that cloud and
fog waters may have contributed to the radioactive contamination of soil and forest ecosystems
(Imamura et al., 2020). This was experimentally demonstrated thanks to field observations at
elevated sites that experienced cloudy conditions in Japan during the nuclear accident (Hososhima
and Kaneyasu, 2015; Kaneyasu et al., 2022).
Subject to the capabilities of trace level measurement, Masson et al. (2015) found that the detection
of $^{134}$Cs (a radionuclide released during the accident with a half-life of 2.06 year) was possible
over a longer time scale in cloud water, compared to its detection in aerosols and rainwater. Wet
deposition by fog has triggered attention for their chemical content (e.g., $NH_4^+$, $SO_2^-$), and more
generally as regards the issue of acidified precipitations and forest decline (Hůnová et al., 2011;
Katata, 2014), as well as for their organic content (Eckardt and Schemenauer, 1998; Kaseke et al.,
2018). In light of the various and sometimes severe impacts of fog on the environment, human
activities (e.g., airport traffic disruption) and human health, fog research and studies have a long
rich history (Pérez-Díaz et al., 2017). While there has been growing interest in the study of fog in
arid and semi-arid regions (Eckardt and Schemenauer, 1998; Feigenwinter et al., 2020; Katata et
al., 2010; Spirig et al., 2021), research on fog deposition and impact on radionuclides settlement
in such environments is notably lacking.

Similarly to cloud water, fog water is also prone to be enriched in atmospheric chemicals or
radioactive compounds. Deposition velocity of fog water droplets depends on both the droplet size
(up to several tens µm) and turbulent movements induced by obstacles (canopy, plant density,
roughness scale, foliar index) and wind components (Tav et al., 2018). Tav et al. (2018) showed
that the fog droplet deposition velocity was similar to the gravitational settling velocity on bare
soils but was systematically higher above short plants and grass. Additionally, rainwater
monitoring networks (i.e, rain gauges) and rain radars are not sensitive enough to quantify the
amount of fogwater deposited. As a result, fog deposition is often called "occult deposition". Once
deposited, water will evaporate and the dry solute will stay on the deposition surface. From the
modeling point of view, and to a few exceptions (Katata, 2014), radionuclide deposition during
fog and cloud events is usually not taken into account in emergency situations response models or
even less in routine conditions.



Fog is a common occurrence in the United Arab Emirates (UAE), which is situated on the northeastern coast of the Arabian Peninsula. Despite being an arid country with desert as the main land cover type (Fonseca et al., 2023; Francis et al., 2021; Nelli et al., 2022), the majority of low visibility events (< 1 km) during winter in the UAE are due to condensation processes (Aldababseh and Temimi, 2017). Fog occurs mostly in winter, with up to 13 fog days per month and 51 fog days per year reported at the international airport in Abu Dhabi (Mohan et al., 2020); (Weston and Temimi, 2020; Weston et al., 2021a). The most common fog type is radiation fog, which depends on the land-sea breeze for its formation (Fonseca et al., 2023; Mohan et al., 2020; Weston and Temimi, 2020; Weston et al., 2021a). Most events last 1 hour or less, but events longer than 9 hours have been observed in January and December (Fonseca et al., 2023; Mohan et al., 2020; Weston and Temimi, 2020; Weston et al., 2021a). Fonseca et al. (2023) examined the atmospheric circulation patterns that favor the occurrence of long-lasting fog events in the UAE, and found a positive trend in the frequency of such occurrences in recent decades.

Although previous analyses of fog in the UAE have been conducted using satellite data (Weston and Temimi, 2020a) and in-situ measurements for a single fog event (Weston et al., 2022), a comprehensive observational analysis of fog microphysics and dynamics in the region has not yet been carried out. This paper addresses this gap by examining the dynamics and microphysics of 12 fog events during the 2021-2022 winter season to determine fog deposition velocity, a crucial parameter for atmospheric dispersion and deposition applications.

Fog deposition velocity plays a significant role in estimating pollutant and radionuclide deposition rates in the environment. However, this parameter has not been estimated for the UAE, despite the country experiencing an average of 50 days of radiation fog per year. Besides allowing for an improvement of the understanding of fog dynamics, an estimation of the fog deposition velocity is crucial for specific applications such as dispersion and deposition models and air quality. Regarding the former, and as noted by Imamura et al. (2020), the ground deposition of radioactive material may be increased in the presence of fog owing to the potential entrainment of the radionuclides into the water droplets and their subsequent gravitational sedimentation to the surface. Incorporating deposition velocity in foggy conditions can therefore improve the accuracy and representativeness of radionuclide dispersion and deposition.

The objectives of this study are as follows: (1) Present the first observation-derived fog deposition velocity for the UAE, (2) Assess, through numerical modeling, the impact of fog on radionuclides dispersion for a given case study. The paper is organized as follows: in section 2, the data and methods used in this study are described. In section 3, we analyze the fog microphysics and dynamics. Section 4 details the fog deposition velocity analysis. Section 5 addresses the impact of fog on the dispersion and deposition of radionuclides in the UAE for a given case study. Conclusions are drawn in section 6.



## 2. Data and Methods

### 2.1 Site Description

The Barakah Nuclear Power Plant (BNPP) is situated in a low-lying coastal region in the UAE's hyperarid western region. According to Nesterov et al. (2021), the site experiences a seasonal mean local sea surface temperature (SST) ranging from about 21°C in winter to 33°C in summer, which is slightly lower than SSTs further east around Abu Dhabi and Dubai. This creates a steeper land-sea temperature gradient, resulting in stronger sea-breeze circulations at the site. Nelli et al. (2022) reported that the wind speeds at the site of interest are generally below 5 ms$^{-1}$, blowing mainly from a northerly direction. The wind intensifies at around 12–18 Local Time (LT) and 02–09 LT due to the sea-breeze and the downward mixing of momentum from the nighttime low-level jet, respectively. Winter sees stronger winds, which vary mostly in response to mid-latitude baroclinic systems, while summer experiences more quiescent conditions due to the site's proximity to the core of the Arabian Heat Low (AHL) (Fonseca et al., 2022). The colder nights and weaker wind speeds from December to February are favorable for radiation fog occurrence, with peak fog formation around local sunrise. Organized convective systems lead to deep and very deep convective clouds in March-April, with generally reduced cloud cover from May to October (Nelli et al., 2021b). The region of interest experiences a monthly-mean aerosol optical depth ranging from ~0.3 in December-January to ~1.2 in July due to increased exposure to dust storms in the summer season. Dust activity peaks during winter and spring associated with the intrusion of cold fronts from mid-latitudes, as reported by Nelli et al. (2020b, 2022).

### 2.2 Field experiment setup

#### 2.2.1 Horizontal Visibility

A SENTRY™ Visibility Sensor 1 (SVS1) instrument was used to measure visibility. The instrument estimates the scattering of visible light by the atmosphere to calculate the extinction coefficient, μ, which represents the amount of attenuation of a beam due to scattering and absorption by aerosols. Horizontal visibility data is recorded every minute. Visibility data from SVS1 are validated and extensively used for detecting occurrences of fog and dust at the Abu Dhabi location (Nelli et al., 2022; Temimi et al., 2020a; Weston et al., 2022).

#### 2.2.2 Fog Monitor

We used a Fog Monitor (FM-120) from Droplet Measurement Technologies to observe and quantify microphysical processes within the fog. This instrument has been largely used in the field and proven to be reliable. Nevertheless, this deployment, to the authors' knowledge, is the first in the region, which allows for comparison with other studies (Gonser et al., 2012; Gultepe et al., 2009, 2021; Mazoyer et al., 2019; Niu et al., 2010). The FM-120 is a forward scattering spectrometer probe that calculates droplet size based on scattered light from a laser and employs



Mie theory. The instrument assumes that the droplets are spherical and made of water with a known
refractive index. The FM-120 can count droplets diameter in the size range of 2-50 $\mu m$, with size
bins of 1 µm for droplets <= 14 $\mu m$ and size bins of 2 µm for droplets > 14 $\mu m$ and <= 50 $\mu m$. The
liquid water content (LWC) is calculated based on the assumption that each droplet is spherical.
Data was recorded every second and later aggregated into 1-minute averages. When processing
median volume diameter (MVD) data, we only considered values where the number concentration
was greater than 2 counts $cm^{-3} sec^{-1}$

### 2.2.3 Microwave Radiometer

We deployed a ground-based passive microwave radiometer (MWR, RPG-HATPRO; G5
series) from RPG at the site to measure brightness temperature across 14 channels. The instrument
has seven channels in the range 22.24–31.40 GHz (K-Band), which are used for retrieving water
vapor, and seven channels between 51.26–58 GHz (V-Band), which are used for retrieving
temperature profiles (Rose et al., 2005). The MWR is capable of capturing temperature profiles
with a high vertical resolution (30-50 m) in the lower 1,200 m of the atmosphere and humidity
profiles with a 100 m resolution in the lower 2,000 m. Here, we analyze the temperature and
specific humidity profiles during the fog events. The full description of the operational method
and measurement uncertainties are discussed in Temimi et al. (2020a). MWR instrument data over
UAE has been extensively used to study the thermodynamic structure of the atmosphere during
different weather conditions such as fog, dust storms, cold fronts, and solar eclipses (Nelli et al.,
2022, 2021a; Temimi et al., 2020a; Weston et al., 2021b).

### 2.2.4 Weather Stations

A portable weather station equipped with the WS501-UMB smart sensor and SW100 smart
disdrometer from LUFT is used to measure meteorological parameters at the site. The parameters
measured by this weather station include air temperature, relative humidity, wind speed and
direction, global shortwave radiation, precipitation type, and precipitation amount. All parameters
are logged at a 1-minute time resolution.
The experiment site is located within 2 km of the Arabian Gulf in a developing urban
region. All the instruments described above, except for the 3D ultrasonic anemometer, were
deployed on the rooftop of the Visitor's Badging Office (VBO, 23.968052°N, 52.267309°E),
which is roughly 10 m above the ground level. Since this building is far away from major built
structures, the measurements are unlikely to be affected by their deployment on its rooftop. The
collected measurements and their respective periods are summarized in Table 1. The 3D ultrasonic
anemometer is installed at 4-m height on a 10-m high meteorological tower located near the beach
area, 23.96333°N, 52.21185°E (Francis et al., 2023; Nelli et al., 2022). The radial distance between
the 10-m tower and VBO is approximately 6 km. An overview of the site where the instruments
are deployed is given in Nelli et al. (2022). Prior to the present analysis, we applied rigorous quality



control procedures described in Rao and Narendra Reddy (2018), Rao and Reddy (2019) to the
measurements, which included removing outliers and performing double coordinate rotation on
the 3D ultrasonic anemometer data etc.

| (a) | (b) |
|---|---|

WRF nested domain setup around BNPP

D01

D02

D03

BNPP

km
200

model terrain height (m)

(c) Overview of instrumentation deployed at Barakah, United Arab Emirates




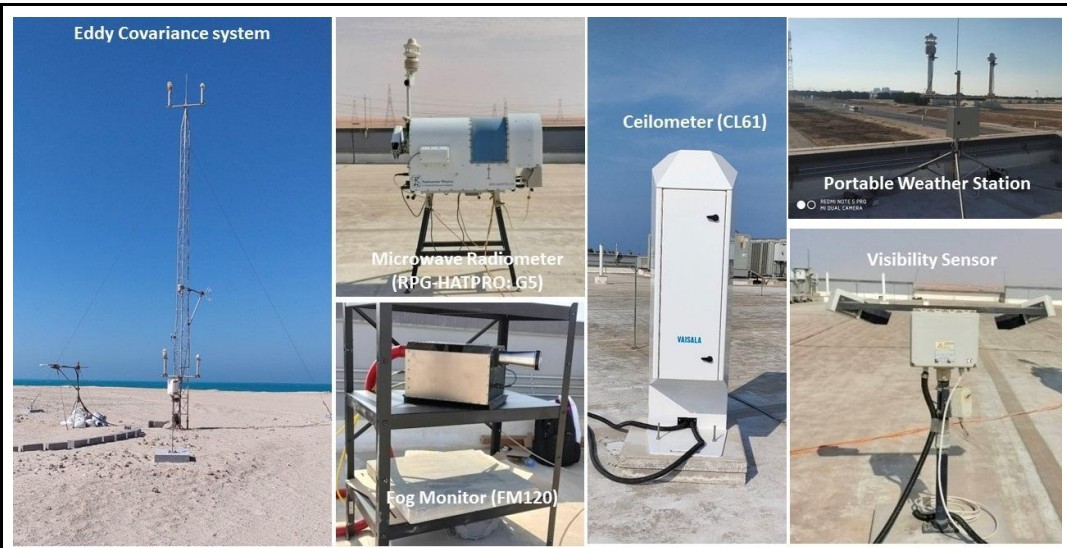

**Figure 1:** (a) WRF domain configuration. The star symbol depicts the BNPP location. The outermost boundaries denote the parent domain (D01, 25 km). D02 (5 km) and D03 (1 km) are the nested domains. (b) Zoom-in view of the innermost domain (D03) with shading giving model orography (m). The location of the BNPP is depicted by a star symbol. (c) Overview of instrumentation deployed at Barakah, United Arab Emirates.

184
185

**Table 1:** List of instrumentation collocated at Barakah nuclear power plant, parameters analyzed, and periods of data analysis.

| Instrument (Make and Model) | Analyzed variables | Analysis period |
|---|---|---|
| Visibility (SentryTM, 73000S) | Horizontal visibility | 27 January - 27 April 2021 |
| Fog monitor (Droplet Measurement Technologies, FM-120) | Liquid water content and particle size distribution | 18 October 2021 - 16 March 2022 |
| Portable weather station (LUFT, WS501-UMB smart weather sensor, WS100 smart disdrometer) | Air temperature, relative humidity, wind speed and wind direction | |
| Microwave radiometer (RPG-HATPRO; G5 series) | Air temperature and relative humidity profiles | |
| 3D Ultrasonic anemometer | Three components of wind (u, v, w) | 9 December 2021 - 25 May 2022 |



### 2.3 Satellite and Reanalysis Products

Geostationary satellite images are used to assess the spatial extent of the fog events. For this, the Spinning Enhanced Visible and Infrared Imager (SEVIRI) instrument onboard Meteosat-10 (situated over 9.5° E) was utilized (Schmetz et al., 2002). The resolution is about 4 km and the images are available every 15 minutes. Day and night-time RGB (red, green, blue) images are created based on the EUMETSAT guidelines. The daytime natural color RGB (red, green, blue) product images are created using the visible and near infrared channels as follows: NIR1.6 (red), VIS0.8 (green) and VIS0.6 (blue). In these images, the fog appears as a white cloud during daytime. In the night-time microphysical product, the red channel is the difference between 12.0 and 10.8 $\mu m$ channels (linear stretch − 4 to 2 K), the green channel is the difference between 10.8 and 3.9 $\mu m$ channels (linear stretch 0 to 10 K) and the blue channel is the 10.8 $\mu m$ channel (linear stretch 243 to 293 K). The fog appears as light green in this product. The principle of the night-time RGB is that fog has a lower emissivity in the 3.9 $\mu m$ channel than 10.8 $\mu m$ (Cermak and Bendix, 2007; Ellrod and Gultepe, 2007; Gultepe et al., 2017). This difference is captured in the green channel in the RGB and generally makes it distinguishable from cloud or land surface.

In addition to in-situ measurements and Satellite data, we used ERA-5 reanalysis data from the European Centre for Medium-Range Weather Forecasts (Hersbach et al., 2020) to present the overall synoptic conditions during the fog events. The ERA-5 data set was selected because it offers higher spatial resolution (0.25°, about 27 km) and temporal resolution (hourly) compared to other reanalysis data sets. Additionally, ERA-5 has demonstrated superior performance in the region, as shown in previous studies (Mahto and Mishra, 2019; Tahir et al., 2020; Taraphdar et al., 2021).

### 2.4 Numerical Modeling

#### 2.4.1 Meteorological Model: WRF

In this study, we use the Advanced Research Weather Research Forecasting (ARW) v4.2 model (Skamarock et al., 2008), a largely used community mesoscale model developed by the National Center for Atmospheric Research (NCAR), to simulate meteorological conditions around BNPP. To better resolve local-scale atmospheric circulation around the nuclear site, we use the same WRF model configuration as in Abida et al. (2022). The domain setup consists of three two-way interactive telescoping nested domains centered on BNPP's location (23.9696° N, 52.2359° E) with the respective resolutions of 25 km, 5 km, and 1 km as depicted in Fig. 1a. Forty-five unequally spaced sigma-pressure vertical levels are used with approximately 11 vertical levels below 1 km above ground level to capture the vertical structure of the fog. The innermost domain, representing the study area, consists of $311 \times 311$ grid points and encompasses a region within a radius of 150 km from BNPP. The physical parameterizations schemes used in all model domains include RRTMG for shortwave radiation (Iacono et al., 2008), RRTM for longwave radiation



(Iacono et al., 2008), Kain-Fritsch cumulus (Kain, 2004), thermal diffusion land surface scheme,
and WSM 3-class microphysics (Hong et al., 2006). This microphysics scheme includes three
categories of hydrometeors: water vapor, liquid water (including cloud  water and rain), and frozen
water (including ice cloud and snow) The liquid water components and the frozen water are treated
for temperatures above and below 0°C, respectively.
Regarding planetary boundary layer (PBL) schemes that parametrize turbulent vertical fluxes of
heat, momentum, and moisture in the PBL, we considered three local PBL schemes, MYJ (Janjić,
1994), MYNN2.5 (Nakanishi and Niino, 2006), and MYNN3.0 (Nakanishi and Niino, 2006), and
two nonlocal PBL schemes, YSU (Hong et al., 2006) and ACM2 (Pleim, 2007). Note that for
stable or neutral conditions, the ACM2 scheme shuts off nonlocal transport and uses local closure.
This approach helps to determine the sensitivity of PBL parameterization for fog simulation. It is
worth mentioning that in WRF, some PBL schemes are tightly tailored to particular surface layer
schemes. Table 2 summarizes these PBL schemes with their associated surface layer options.

**Table 2**: WRF used PBL Schemes

| PBL scheme | Closure type | Surface Layer scheme | Short description |
|---|---|---|---|
| MYJ | 1.5 local | Monin-Obukhov (Janjic Eta) scheme | An improved local closure scheme based on Mellor-Yamada 1.5 order local scheme using the turbulent kinetic energy (TKE) model. It is suitable for stable flows, making it more appropriate for fog modeling, but it might underestimate vertical mixing. |
| MYNN2.5 | 1.5 local | Monin-Obukhov (Janjic Eta) scheme | An updated scheme, TKE-based with level 2.5 turbulent closure model. It provides improvements in the surface layer, by modifying the exchange coefficients and non-local terms, resulting in better performance under very stable conditions. It improves the mixing length scale which leads to better control of the inadequate growth of the convective boundary layer. |
| MYNN3.0 | 2.0 local | MYNN | Similar to MYNN2.5 but it runs at level 3.0 closure. With a more accurate representation of the boundary layer, it may show some ability to simulate radiation fog. |
| YSU | 1.0 nonlocal | Revised MM5 Monin-Obukhov scheme | A non-local closure scheme with a counter-gradient flux for heat in the unstable boundary layer. It is often used for its simplicity and computational efficiency. It may produce an excessive vertical mixing during the evening that leads to too deep PBL height. |
| ACM2 | 1.0 nonlocal | Revised MM5 Monin-Obukhov scheme | A non-local closure scheme, which predicts the boundary layer height, with an enhanced mixing for the stable boundary layers. It has a counter-gradient term for heat and moisture, making it well-suited for deep convective situations, but may over deepen the PBL. |




The choice of a PBL parameterization scheme can significantly impact the accuracy of fog
simulation (Chaouch et al., 2017; Chen et al., 2020; Lin et al., 2017; Nelli et al., 2020a; Temimi et
al., 2020b). PBL schemes are classified based on two factors: the order of the turbulence closure
and the use of local or nonlocal mixing approaches. Turbulence closure refers to the mathematical
approach used to represent the effects of small-scale turbulent motions on larger-scale atmospheric
variables. Local closure schemes estimate the turbulent fluxes based on the mean atmospheric
variables at each point on the model grid, or possibly their gradients. In contrast, nonlocal closure
schemes use multiple vertical levels to determine variables at a given point. This approach allows
for the consideration of the effects of turbulence at other levels of the atmosphere.
Meteorological analysis from the National Centre for Environmental Prediction (NCEP) Global
Forecast System (GFS) with a horizontal spatial resolution of 0.25° and with a time resolution of
3h were used to provide initial and boundary conditions to WRF experiments. Besides atmospheric
fields, time-varying analyzed SSTs (Sea Surface Temperature) within NCEP GFS were also
supplied to the model.
**2.4.2 Atmospheric Dispersion Model: FLEXPART**
In this study, we use the Lagrangian particle dispersion model, FLEXPART-WRF (Brioude et
al., 2013),  a widely used model for simulating atmospheric and deposition processes at various
spatial scales, ranging from small-scale dispersion of pollutants from power plant stacks to long-
range transport applications (Bahreini et al., 2009; Cooper et al., 2010). In addition to transport
and turbulent diffusion, the model considers radioactive decay and simulates both dry and wet
deposition of aerosols and gasses. The model works by computing trajectories of a large number
of small parcels of air containing initial concentrations of the species. These air parcels are then
advected downstream, subject to the unsteady velocity field, turbulence characteristics, turbulent
diffusion, as well as dry and wet deposition processes.
We recall that, in FLAXPART, the wet deposition is calculated based on the precipitation rate by
treating separately the in-cloud and below-cloud scavenging processes (Stohl et al., 2005). In
contrast the dry deposition velocity is calculated using the electric resistance method (Seinfeld et
al., 1998). The gravitational settling velocity component in the dry deposition parametrization for
a particle is given by

$$v_g = g\rho_p d_p^2 C_{cun}/18\mu \tag{1}$$

where $\rho_p$ and $d_p^2$ are the particle density (kg m$^{-3}$) and diameter (m), respectively, $\mu$ is the dynamic
viscosity of the air (kg m$^{-1}$ s$^{-1}$), $C_{cun}$ is the Cunningham slip-flow correction, and $g$ is the
acceleration of gravity (9.81 m s$^{-2}$).
Foggy atmospheric conditions can lead to the entrainment of radionuclides in water droplets,
which can then be deposited on earth's surface when the fog comes in contact with it. Additionally,



dry aerosols serve as Cloud Condensation Nuclei and can then grow by condensation of water
vapor (Seinfeld et Pandis, 1998) leading to droplets prone to settle as a result of gravitation and
turbulent impaction on obstacles. The coalescence process doesn't act on the released
radionuclides, but it affects the size distribution of the fog droplets.
To account for the fog deposition processes as an additional scavenging agent in our dispersion
simulation, we slightly modified the gravitational settling in equation 1 by adding to it an
additional thresholding term. In a given model grid cell, if the LWC (averaged over the lower three
model levels) is found to be greater than 0.1 g m$^{-3}$, then this term is set to the value of 3 cm s$^{-1}$, as
a representative value of fog deposition rate around Barakah site. Otherwise, it is set to zero. The
minimal threshold value of 0.1 g m$^{-3}$ to determine whether a grid cell is foggy or not was chosen
based on trial simulations which showed that the WRF model tends to overestimate the near-
surface liquid water content when compared to LWC observations at BNPP (see later, Fig. 11a).
**2.5 Detection of fog over experiment site**
Fog is a meteorological condition where horizontal visibility becomes less than a conventionally
accepted threshold of 1000 m due to the presence of water droplets. In this study, fog events are
detected based on the horizontal visibility measured at a height of 10 m. First, all time-steps where
horizontal visibility is less than or equal to 1 km are labeled as potential fog events (Fonseca et al.,
2023; Mohan et al., 2020; Nelli et al., 2022; Temimi et al., 2020a). Afterwards, a visual inspection
of 15-minute SEVIRI fog RGB images is conducted to confirm the formation of fog on the selected
dates and times. Additionally, cloudy or rainy conditions are filtered out using the 30-min merged
IR brightness temperature data from geostationary satellites (Nelli et al., 2021b; Rao et al., 2013;
Reddy and Rao, 2018). A detailed list of each fog occurrence, including the date, time, duration,
and minimum horizontal visibility during each fog event is given in Table 3.

**Table 3:** Fog occurrence timings (onset, dissipation, and duration) at Barakah and minimum horizontal visibility for each of the 12 fog events.

| S.No. | Start Date | Fog occurrence timings | | | Minimum horizontal visibility (m) |
|---|---|---|---|---|---|
| | | **Onset** | **Dissipation** | **Duration (hour)** | |
| 1 | 2021-01-29 | 04:32 LT | 05:39 LT | 01:01 | 241 |
| 2 | 2021-02-04 | 00:32 LT | 08:03 LT | 02:04 | 154 |
| 3 | 2021-02-05 | 02:35 LT | 08:03 LT | 05:27 | 120 |
| 4 | 2021-02-13 | 06:06 LT | 08:05 LT | 00:31 | 130 |





| 5 | 2021-02-16 | 00:28 LT 06:38 LT | 04:40 LT 08:56 LT | 04:12 02:18 | 90 |
|---|---|---|---|---|---|
| 6 | 2021-02-17 | 00:22 LT | 06:14 LT | 01:55 | 165 |
| 7 | 2021-11-23 | 05:01 LT | 09:10 LT | 03:53 | 102 |
| 8 | 2022-01-27 | 02:43 LT | 08:02 LT | 04:03 | 261 |
| 9 | 2022-02-04 | 02:01 LT | 09:20 LT | 06:15 | 198 |
| 10 | 2022-02-11 | 06:00 LT | 06:31 LT | 00:32 | 200 |
| 11 | 2022-02-24 | 04:06 LT | 09:33 LT | 04:07 | 44 |
| 12 | 2022-02-25 | 00:28 LT | 00:54 LT | 00:13 | 253 |

Microphysics and dynamics for these 12 fog events are presented in sections 3 and 4. In section
5, the impact of the long-lasting fog event on February 16 2021 on ground deposition of radioactive
materials in case of an accidental radioactive release at Barakah Nuclear Power Plant (BNPP) is
presented.
**2.6 Fog deposition flux calculations**
In this section, we describe how the total vertical fog water flux between the surface and the
atmosphere and the fog deposition velocity are estimated using data from the fog monitor sampler
and ultrasonic anemometer installed at BNPP.
Before calculating any quantity related to fog deposition, we first pre-processed fog monitor data
for all recorded fog events at the site. We recall that the fog monitor provides information on LWC,
the number of water droplets per bin droplet size, and the Median Volume Diameter (MVD) every
1 second. The data were re-sampled to 1-minute frequency. The concentration number $N_i(t)$ per
size bin for each time step $t$ ($\#\,\mathrm{cm}^{-3}$) is calculated by
$$N_i(t) = \frac{n_i(t)}{dw_i * SV(t)} \quad (2)$$


where $n_i(t)$ and $SV(t)$ are the count of droplets in size bin $i$ and the sampling volume in $\mathrm{cm}^3$ at
time $t$, respectively; $dw_i$ is the width of the size bin $i$ in logarithmic scale calculated as $dw_i =$
$log(u_i/l_i)$, with $u_i$ and $l_i$ being the upper and lower bounds of the size bin $i$, respectively. Based
on the assumption of spherical shape of fog water droplets, the liquid water content $LWC_i$ (kg m$^{-3}$
$^3$) available within the bin droplet size $i$ at time $t$ is given by:

$$LWC_i(t) = \frac{1}{6}\pi d_i^3 N_i(t) dw_i \rho_w \quad (3)$$



where $\rho_w$ is the density of water (kg m$^{-3}$), $d_i$ is the geometric mean diameter ($\mu m$) for each droplet
size class. As in Tav et al. (2018), we calculated the concentration number density ($\rho_{nc}(i,t)$), and
the mass density ($\rho_{lwc}(i,t)$) of each size bin $i$ and at each time step $t$ as follows:

$$\rho_{nc}(i,t) = \frac{1}{dw_i}\left(\frac{N_i(t)}{\sum_{i=1}^{30} N_i(t)}\right)$$

$$\rho_{lwc}(i,t) = \frac{1}{dw_i}\left(\frac{LWC_i(t)}{\sum_{i=1}^{30} LWC_i(t)}\right)$$

Note that the total water flux in the fog is a combination of the turbulent liquid flux, gravitational
settling of droplets, water condensation and an advection component. Yet, the last two processes
lead to flux magnitudes at least six orders lower than the first two, as reported by Spirig et al.
(2021) and Eugster et al. (2006). In addition, the determination of the liquid water flux due to the
advection would require the installation of several fog monitors around the nuclear site, which is
beyond the scope of this study.
Thus, in this study, the total flux of liquid water in the fog (kg m$^{-2}$ s$^{-1}$), $LWF_{total}$, is expressed only
by the sum of the turbulent and gravitational settling components:
$$LWF_{total} = LWF_{turb} + LWF_{grav} \qquad (4)$$
The gravitational settling flux, $LWF_{grav}$ is calculated based on Stoke's law for settling velocity
(Beswick et al., 1991):

$$v_s = gd^2(\rho_{water} - \rho_{air})/18\eta_{air} \quad (5)$$

where $g$ is the gravity acceleration ($9.81$ m s$^{-2}$), $\eta_{air}$ is the dynamic viscosity of the air (kg m$^{-1}$ s$^{-1}$),
$d$ is the droplet diameter ($\mu m$), $\rho_{water}$ and $\rho_{air}$ are respectively the water and air densities (kg m$^{-3}$
$^{3}$). The settling velocity $v_s$ (m s$^{-1}$) is calculated for each droplet size class $i$ and multiplied by the
liquid water content within this size class, $LWC_i$. The total gravitational settling flux is then the
sum of the interval fluxes over all droplet sizes:

$$LWF_{grav} = \sum_i - v_{s,i} LWC_i \quad (6)$$

Note that the gravitational settling process is more efficient for water droplets of larger sizes,
typically greater than $10\ \mu m$.




In contrast, the turbulent deposition process which depends on the turbulent exchange between the
atmosphere and the surface, is more effective for small water droplets because they can more easily
follow the chaotic, turbulent flow of the air and remain suspended in it for longer periods of time.
The liquid water turbulent flux $LWF_{turb}$ is calculated based on the Eddy Covariance (EC) technique.
First, a double rotation correction was used to align the wind horizontal component ($u$) with the
mean wind to obtain a zero average for the other wind components $v$ and $w$ (Rao and Reddy, 2019).
The EC framework, in which co-located measurements from a fog monitor sampler and a sonic
anemometer are combined, is widely used to derive the turbulent liquid water flux (Beswick et al.,
1991; Degefie et al., 2015; Klemm and Wrzesinsky, 2007; Kowalski and Vong, 1999; Spirig et
al., 2021; Thalmann et al., 2002; Westbeld et al., 2009).

The turbulent vertical flux is subsequently determined as the covariance between the vertical wind
speed ($w$; m s$^{-1}$) and $LWC$ time series:

$$LWF_{turb} = \overline{w'LWC'} \qquad (7)$$


The overbar indicates a 15 min temporal average and the prime indicates the deviation of each
component from its 15 min mean.

The time-dependant settling velocity $v_s$ is calculated as the ratio of the gravitational settling flux
to the 15 min averages of liquid water content:

$$v_s = -\frac{LWF_{grav}}{\overline{LWC}} \quad (8)$$


Note that we can also estimate this quantity directly by applying Stoke's sedimentation velocity
formula using as the droplet diameter size the median diameter volume (MDV) data from the fog
monitor. Both approaches have been found to lead to very similar results (results not shown).

Similarly, the fog droplet net deposition velocity, $v_d$, is determined by calculating the ratio of the
total fog water flux to the 15-min averages of liquid water content using the following equation:

$$v_d = -\frac{LWF_{total}}{\overline{LWC}} \quad (9)$$


Note that a positive sign of $v_d$ indicates a downward flux. It should be noted that a net gain of
water deposition on the surface is not necessarily all-time guaranteed despite the presence of fog.
Tav et al. (2018) by using a weighing measurement setup found that among other factors, the
deposited fog water depends significantly on the type of surface.





**3.    Fog Microphysics and dynamics**
**3.1 Fog microphysics**
The microphysics of fog were analyzed based on observations of 12 fog events that occurred
during the winter seasons of 2021 and 2022. Diurnal variations in horizontal visibility, liquid water
content, and number concentrations were studied for each of the 12 fog events, as depicted in Fig.
395 2.

The onset and maturation of fog at Barakah were found to occur between 0-4 LT and 6-7 LT,
respectively, as shown in Table 3 and in Fig. 2a. Our results are consistent with the diurnal
variations in fog occurrence reported by Nelli et al. (2022) at the Barakah site. Nelli et al. (2022)
used long-term meteorological data from a weather station for their study. Additionally, our
findings are in line with Weston et al. (2021a) report on fog occurrence at Abu Dhabi International
Airport, which is a coastal location in the UAE. Weston et al. (2021a) analyzed the meteorological
aerodrome report (METAR) data to arrive at their conclusions. Interestingly, the onset of fog at
Barakah is more swift than that in fog events at Abu Dhabi (Temimi et al., 2020), both being
coastal sites. Perhaps the radiative cooling is more pronounced given the more arid landscape as
opposed to the urban environment in Abu Dhabi. Regarding the fog microphysical properties, we
observed that, similar to the abrupt decrease in horizontal visibility, both the LWC and number
concentration also increase abruptly. The mean and median of LWC when the horizontal visibility
reduces to less than or equal to 1 km are 150 and 170 $\mathrm{mg\,m^{-3}}$, respectively. Interestingly, the LWC
and number concentrations for the 04 February 2022 event are relatively lower than those for the
16 February 2021 fog event, despite a similar fog duration. This difference is further discussed in
section 3.2, along with the thermodynamics parameters obtained from weather stations and the
measurements collected by the microwave radiometer.

(a)





(b)

(c)



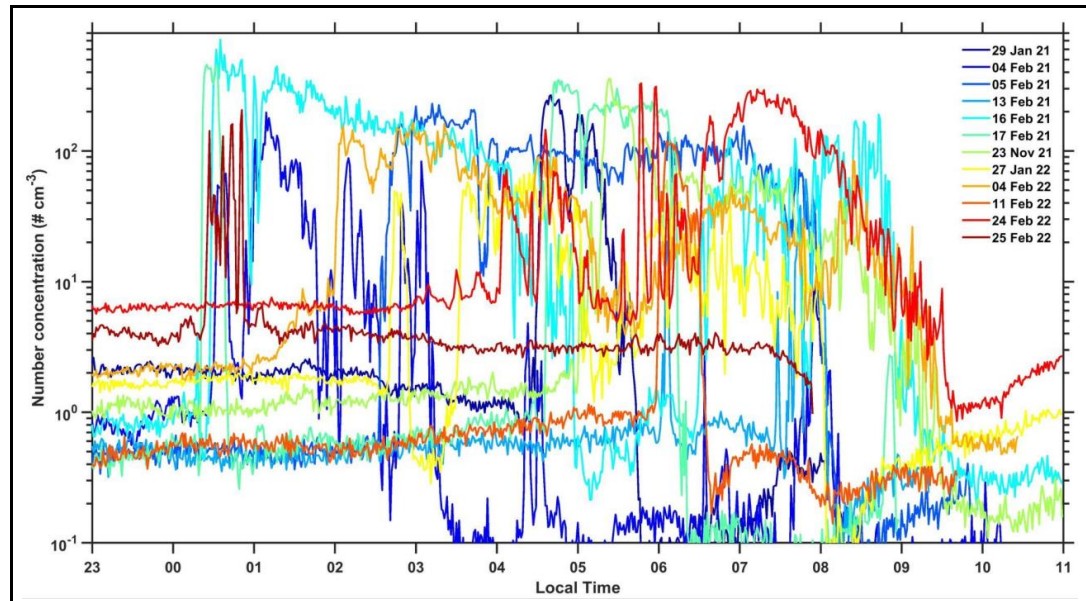

**Figure 2: Characteristics of 12 fog events.** Diurnal variations in (a) horizontal visibility (km), (b) liquid water content (LWC; mg m$^{-3}$) and (c) number concentration of cloud droplets (cm$^{-3}$) for 12 fog events at Barakah.



To better understand the droplet size distribution (DSD), the number concentration (# cm$^{-3}$,
hereafter, $N_c$) in each bin droplet size is averaged over the duration of each fog event and
normalized by the width of the size bin in the logarithmic scale. Fig. 3a indicates that all the fog
events exhibit the same shape of a bimodal distribution, with peaks about 4.5 and 23 $\mu m$. The DSD
of the composite of all fog data with visibility less than or equal to 1 km also shares the same
shape. The ratio of the number concentration in the first mode to the second mode is almost 5,
indicating that the proportion of small droplets is significantly larger than the proportion of large-
size droplets in the fog.

Bimodal distributions of fog water droplet sizes are common as reported by numerous studies in
the literature (Gultepe and Milbrandt, 2007; Westbeld et al., 2009; Weston et al., 2022; Ghude et
al., 2023). Note that the shape of the DSD could also be, in part, influenced by the type of the
condensation nuclei available at this semi-arid coastal site. The bimodal distribution is a result of
the coexistence of two distinct populations of particles. The first mode, 4.5 $\mu m$ corresponds to
smaller droplets which are formed through condensation of water vapor during the initial stage of
the fog. The second mode, 23 $\mu m$ corresponds to larger droplets which are mainly formed through
the coalescence and collision of smaller droplets during the intermediate and mature stages of the
fog as addressed in the subsequent sections. Interestingly, the bimodal distribution is also observed
in LWC distribution as a function of droplet sizes with a more broadening mode at 4.5 $\mu m$ and a



marked peak at 25 $\mu m$ as highlighted in Fig. 3b. The DSD mode at 4.5 $\mu m$ has a very minor
influence on the LWC and most likely formed at the onset stage of the fog. This result reveals that
despite the large proportion of small particles with diameters less than 5.5 $\mu m$ in the fog, the largest
contribution to the LWC originates essentially from medium to large-size droplets between 10 and
35 $\mu m$. The twelve fog cases reveal clear differences between them in terms of the variability of
the LWC and $N_c$ values (Fig. 4a), even though almost all of them are of the radiation fog type.
LWC and $N_c$ range from 10 to 600 mg m$^{-3}$ and 10 to 480 # cm$^{-3}$ respectively, while MVD ranges
from 2.11 to 30 $\mu m$ (Fig. 4b). Overall, and especially for long-lasting fog events (longer than 90
min), we observe that the maximum values for $N_c$, LWC and MVD obtained in this study tend to
be higher compared to other values reported in the literature, in particular for the values reported
for continental fog events (Boutle et al., 2018; Egli et al., 2015; Formenti et al., 2019; Gultepe and
Milbrandt, 2007; Mazoyer et al., 2019). However, it should be noted that these ranges are sensitive
to the temporal frequency at which the data is re-sampled (1 min in this study).
In addition, the analysis of the spread of MVD data for each fog event, as shown by the box plots
in Fig. 4b, indicates that on average the MVD ranges from 20 to 26 $\mu m$ in almost all the cases,
except for the fog event observed on February 25, 2022, which has the smallest median MVD of
about 5 $\mu m$. This fog event has the shortest lasting time of 13 minutes among all other fog events
and based on SEVIRI Fog RGB images (Fig. A3), is most likely a remnant from a fog formed over
the sea and advected by the winds toward the site when it was detected. Interestingly, we notice
that the fog event recorded on 16 February 2022 shows the lowest variability amongst all other
fog cases with a minimum interquartile range (IQR) of 2.5 $\mu m$ and median MVD of 24 $\mu m$. We
observe that the lower and upper quartiles for most fog cases are between 20 and 28 $\mu m$. This
suggests that radiation fogs occuring during the winter at this hyperarid coastal site tend to have
larger water droplets. Note that the excess of the moisture content and the warm air available at
the site also enhance the growth in size of the droplets, as warm air can hold more moisture.
The relationship between the LWC and $N_c$ in fog is complex and depends on various factors,
including ambient temperature, humidity, and the quantities of aerosol particles that can serve as
nuclei for fog droplet formation. In general, when the LWC increases, the $N_c$ also increases.
However, at some point, the $N_c$ might start to decrease due to coalescence resulting in larger
droplets, which in turn may decrease in number due to gravitational settling. Also, this relationship
can vary depending on the droplet size distribution. To shed some light on this relationship we first
examined the temporal variation of LWC as a function of the droplet sizes, and calculated the
probability density functions of LWC and $N_c$ for the twelve fog events. It was found that the results
for almost all fog events are very similar to each other and therefore lead to the same primary
conclusions. Thus, we only present here the results of the fog event observed on February 16,
2021, which is an interesting case with a longer duration and associated with a greater reduction
in horizontal visibility (see Table 3). Figure 5a shows the time-variation of LWC per size bin as
well as the integrated LWC. The fog event on 16 February 2021 started at 00h28 (LT) and lasted





4:12 hours till 04h40 (LT). The LWC then decreased drastically for about 1.5 hours before starting
to increase when a second fog patch moved in at 06:38 (LT) and lasted for 2:18 hours. The
colormap indicates that LWC reaches its maximum values during the fog maturation phase for
droplet sizes between 20 $\mu m$ and 30 $\mu m$. This again corroborates that the larger-size droplets in
the fog have the largest contribution to the LWC compared to the smaller-size droplets.

(a)

(b)



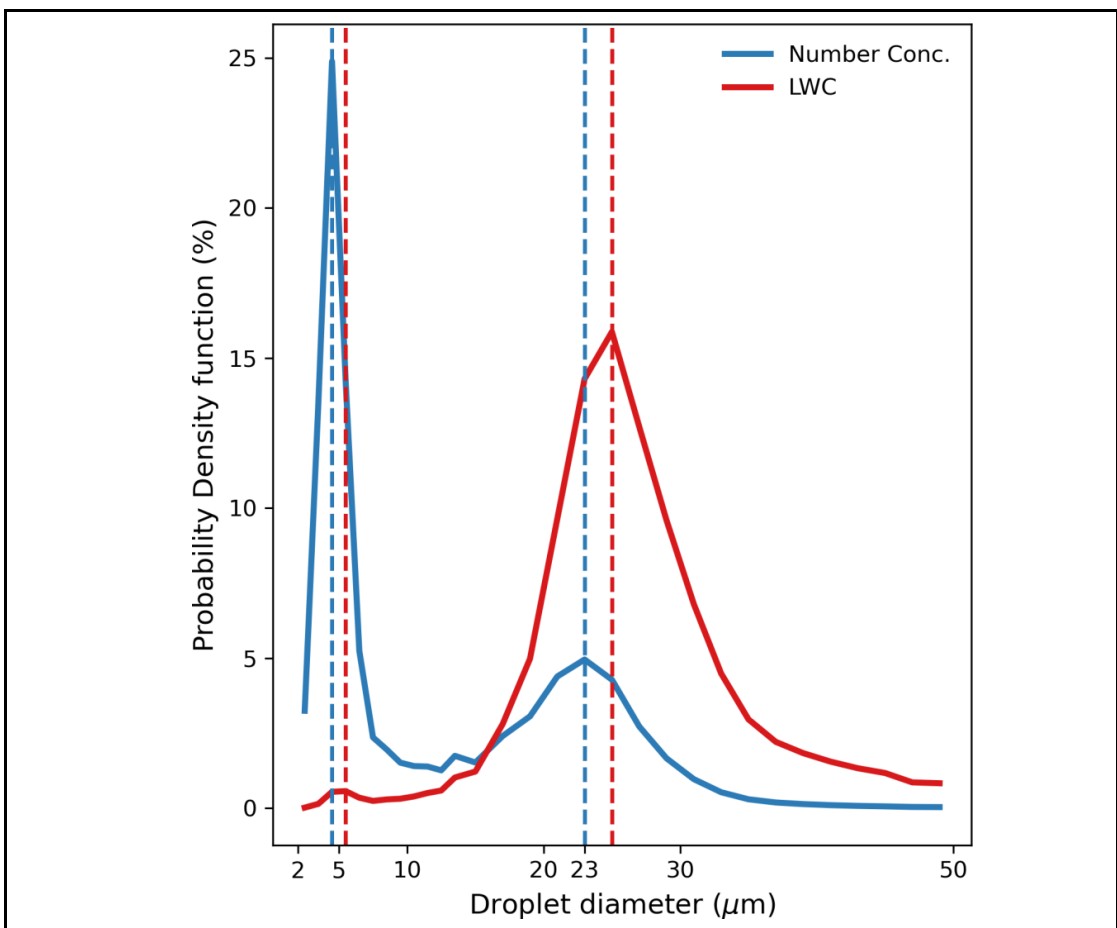

**Figure 3: Microphysics of 12 fog events**. (a) Fog water droplet size distribution for each fog event. The $N_c$ in each droplet size bin (#/cm$^3$/ $\mu m$) is averaged over the duration of each fog event and normalized by the width of the size bin in logarithmic scale ($dN_c/dLogD$). (b) Probability density functions of $N_c$ (blue) and LWC (red) as a function of droplet size.













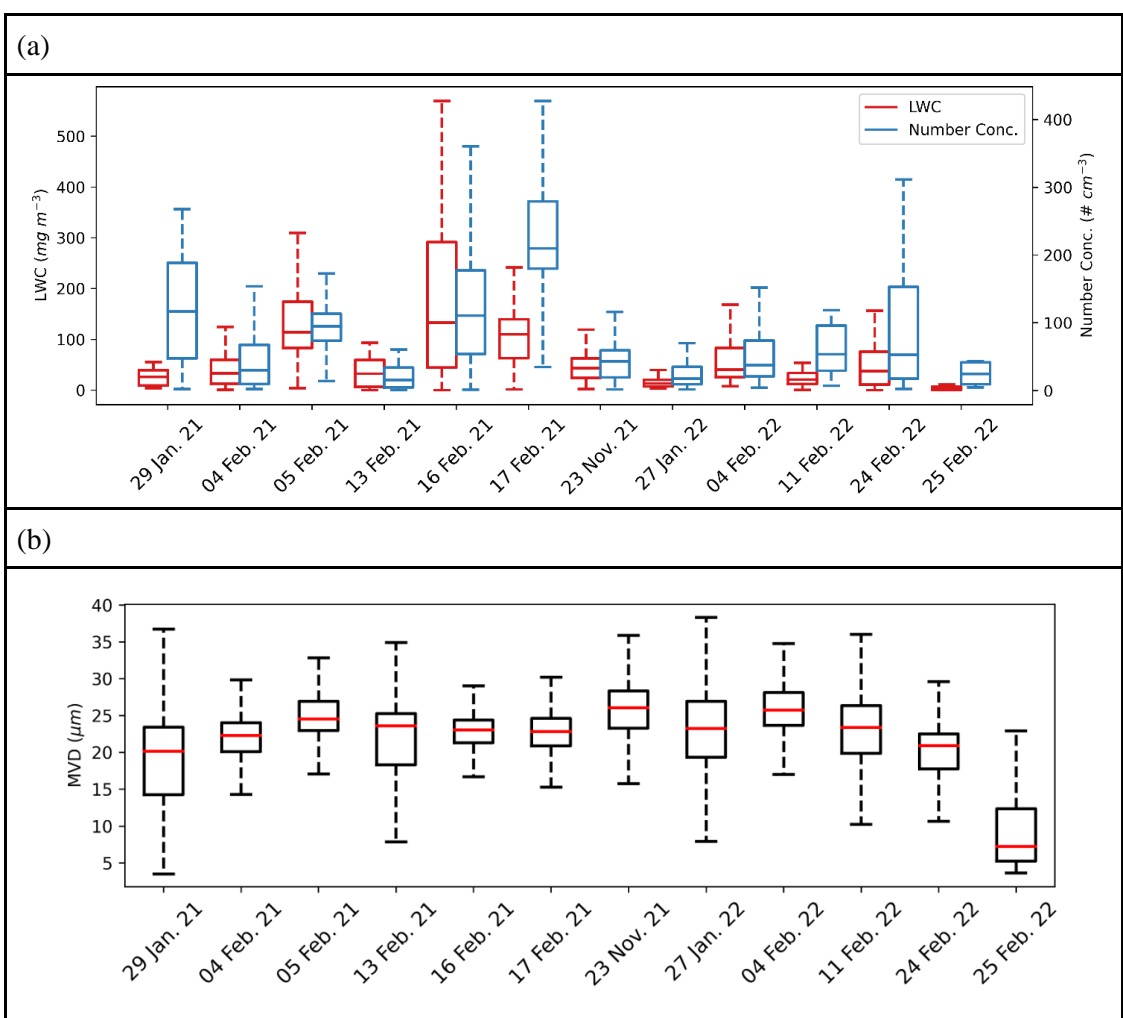

**Figure 4: Statistics of 12 fog events**. Box plots for (a) LWC (red; $mg\,m^{-3}$) and $N_c$ (blue; $\#\,cm^{-3}$) and (b) Median Volume Diameter (MVD; m). Each box plot shows the minimum, median, maximum, and first and third quartiles. The fog monitor data is resampled at 1 min frequency.



Figure 5b shows calculated mass and number probability densities for four typical droplet size
ranges: 1-5 $\mu m$, 5-10 $\mu m$, 10-20 $\mu m$ and 20-50 $\mu m$, which will determine the contribution of each
droplet size class to the total number concentration per $cm^3$ and to the total mass of liquid water
per $m^3$. This will help to explore the size distribution of fog droplets and its contribution to the
liquid water content. Interestingly, we observe that at the onset of the fog, the droplet sizes between
1 and 5 $\mu m$ and between 5 and 10 $\mu m$ represent respectively 60-80% and 20-40% of the total
number of droplets, and between 40 to 60 % of the total mass of liquid water. This suggests that
the LWC is more controlled by smaller to medium droplets at the onset phase. In addition, during





the mature stage, we notice that the percentage of droplets between 1 and 5 $\mu m$ of the total number
density gradually decreases until about 03:15 (LT) when it begins to increase. However, the
opposite is observed for larger droplets particularly between 20 and 50 $\mu m$. Note that large droplets
represent on average less than 20% of the total number of droplet density. This indicates that, at
the onset and mature stages of the fog, the proportion of smaller droplets is significantly higher
than that of larger droplets. The number density of droplets between 5 and 10 $\mu m$ does not change
significantly and fluctuates on average mostly between 20 and 25% in the mature stage. This
reveals that during the mature phase of the fog, the larger droplets are formed mainly by collision-
coalescence process which is more efficient for smaller droplets with sizes between 1 and 5 $\mu m$.
Once formed, the larger droplets can settle to the surface due to gravitational sedimentation and
turbulence, reducing the liquid water content in the air. Combined, droplets between 10 and 20
$\mu m$ and 20 and 50 $\mu m$ account for up to 90% of the condensed water in the air. In particular, the
very large droplets (20 and 50 $\mu m$) alone represent up to 80% of the total mass in the mature stage.
This supports the claim that higher LWC values are primarily attributed to the presence of a
sufficient number density (between 20-40%) of large droplets in the fog. This result is in perfect
agreement with Tav et al. (2018), who studied fog events in France and came to the same
conclusion.

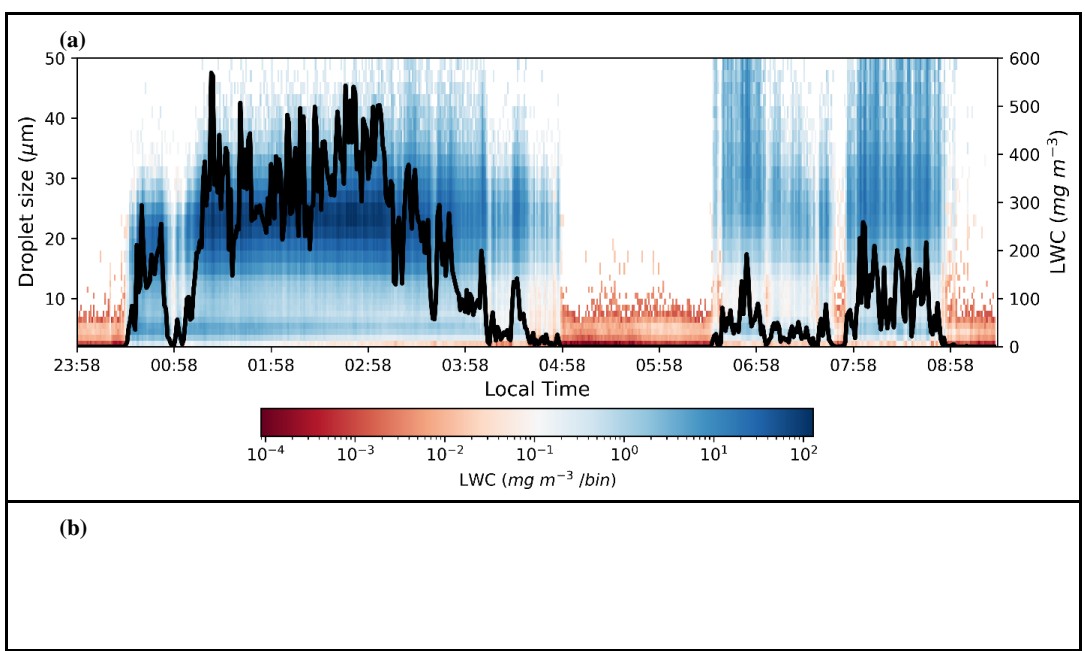

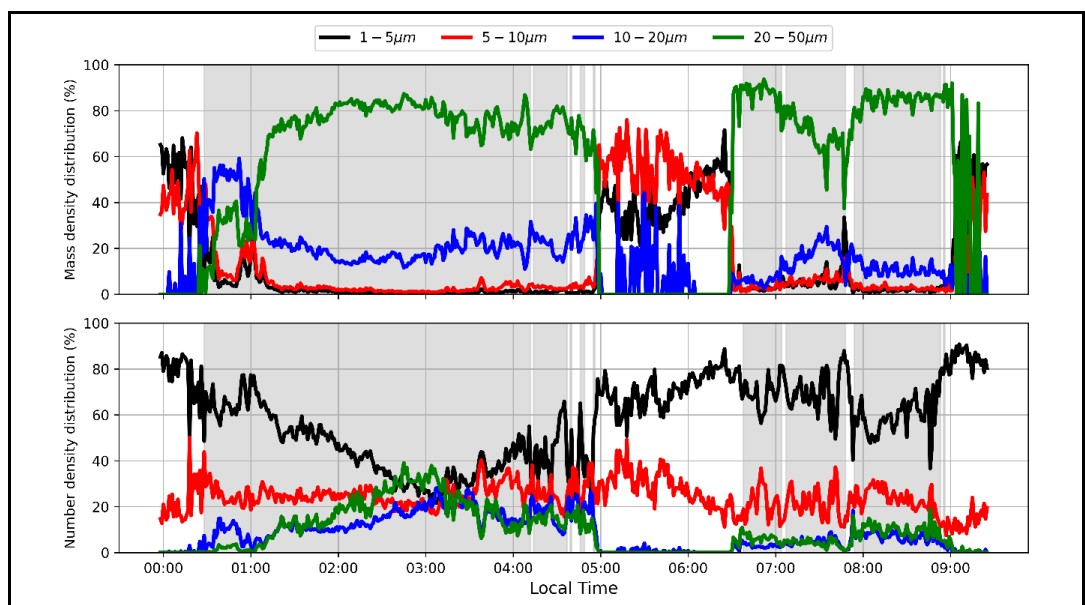

**Figure 5: Microphysics of the 16 February 2021 fog event.** (a) Time-variation of the distribution of LWC as a function of droplet sizes (shading; $mg\,m^{-3}$/bin) and the integrated LWC (black line; $mg\,m^{-3}$). (b) Mass (LWC) and number concentration densities calculated for four droplet-size ranges: 1-5 $\mu m$ (black), 5-10 $\mu m$ (red), 10-20$\mu m$ (blue) and 20-50 $\mu m$ (green). All quantities are calculated using 1-min resampling data of the fog event observed on 16 February 2021.


**3.2 Atmospheric thermodynamics during fog events**
The thermodynamics and dynamics of fog are analyzed based on collocated weather station and
microwave radiometer data. Diurnal variations in relative humidity and wind speed are studied for
each of the 12 fog events, as depicted in Figs. 6a-b. The mean relative humidity and wind speed at
the onset of fog for the 12 cases are 94% and $3\,m\,s^{-1}$, respectively, which are consistent with the
threshold values used by Nelli et al. (2022) for detecting fog occurrence over Barakah using
weather station data. For the fog events on 16 February 2021 and 04 February 2022 the wind speed
exceeded $3\,m\,s^{-1}$ in the middle of the fog event. During this time, an increase in horizontal visibility
(> 1 km) and a sudden decrease in LWC and number concentrations were measured Figs. 2a-c.
To investigate the factors that contribute to the differences in LWC and number concentration
between the February 2021 and 2022 fog events, we analyzed the temperature and specific
humidity profiles retrieved from the microwave radiometer data. Fig. 6c shows that the February
2022 fog events were associated with double temperature inversions on 04 and 24 February, and
a strong temperature inversion on 27 January. Double temperature inversions, similar to those
observed during the February 2022 fog events, were also detected in the ceilometer backscatter
profiles (Francis et al., 2023) prior to the onset of fog (elevated second layer) (Fig. A2). On the





other hand, the 16 February 2021 fog event occurred in a relatively less stable atmosphere with a
single temperature inversion. The strength and depth of temperature inversion for all 12 fog events
are shown in Fig. A1. It is concluded that the inversion depth and strength play a crucial role in
determining the moisture pumping to the fog layer and hence the growth of new droplets, which
affects the liquid water content. Other factors, such as the advection of air mass and descending of
stratus, can also influence the LWC and $N_c$. However, the differences in the temperature and
humidity profiles, as well as the strength and depth of temperature inversion, are likely the primary
contributors to the observed differences in LWC and $N_c$ between the February 2021 and 2022 fog
events. It would be interesting to further investigate the roles of these meteorological factors in fog
formation and evolution using more detailed observations and modeling simulations in future
studies. The vertical velocity at 700 hPa obtained from ERA-5 data (not shown) indicates descent
on 04 and 24 February and weak ascent on 27 January 2022, which also supports the influence of
large-scale descent motion on the fog formation and growth. However, more detailed analyses of
the atmospheric dynamics and thermodynamics are needed to fully understand the complex
interplay between various meteorological factors, aerosols composition and fog properties.

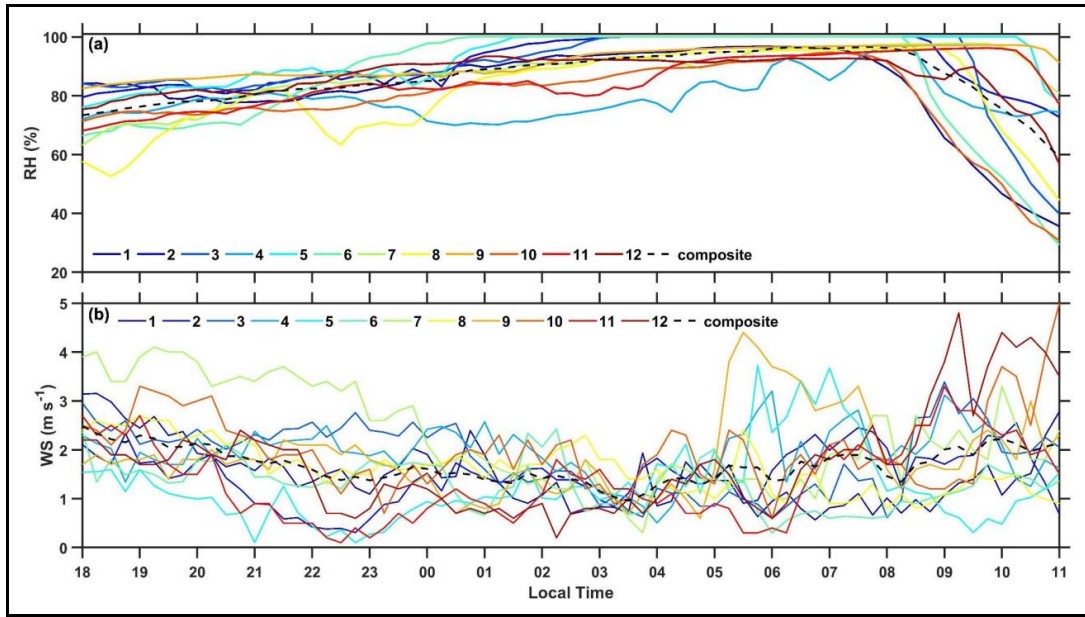



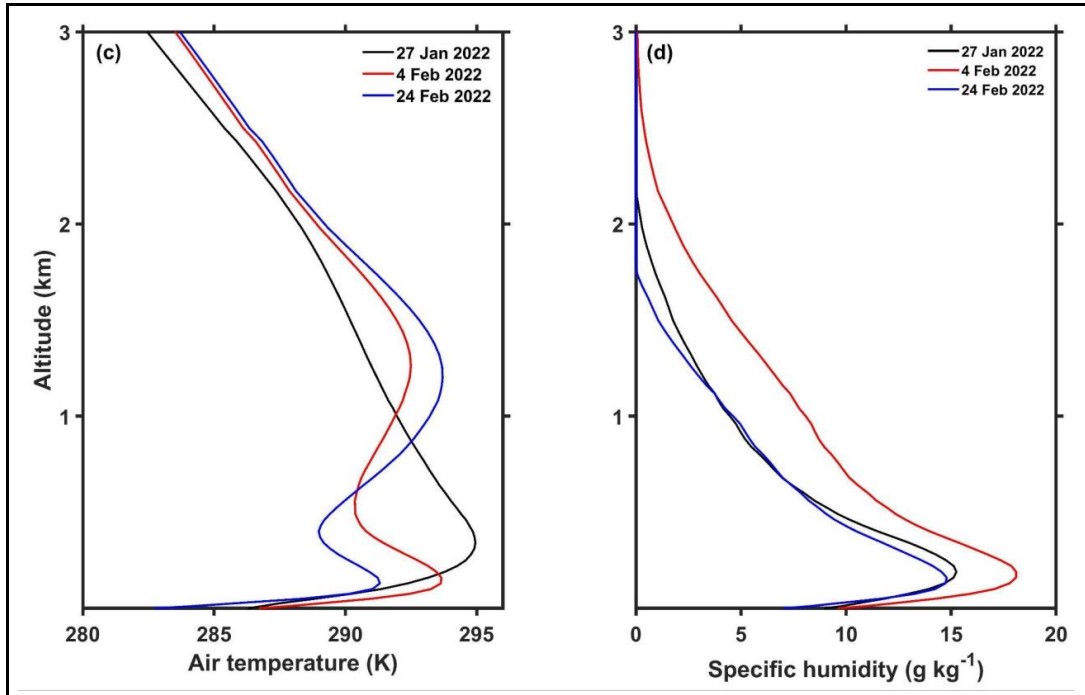

**Figure 6: Atmospheric state during 12 fog events.** (a-b) Diurnal variations (18 LT - 11 LT) in relative humidity (RH; %) and wind speed (m s$^{-1}$) for all fog events. (c) Averaged profiles of temperature (K) from the Microwave Radiometer (MWR) over the duration of 27 January (black), 04 February (red) and 24 February (blue) 2022 fog events. (d) is as (c) but for the specific humidity (g kg$^{-1}$).


An analysis of the large-scale circulation patterns for the 12 fog cases, based on the ERA-5 data,
indicates the following: (i) a low-pressure system over the eastern Mediterranean extending into
Iraq and northern Saudi Arabia; (ii) a ridge over the southeastern Arabian Peninsula; (iii) a strong
southwesterly flow between the two, advecting moisture from the Red Sea into Iran and the
northern Arabian Gulf; (iv) some of the moisture in the northern Gulf is brought into the UAE by
the counterclockwise circulation of the ridge; and (v) weak near-surface wind speeds that favor
the occurrence of fog (Fig. A4). The anomalies show the advection of drier air from continental
Asia into the southeastern Arabian Peninsula (drier air aloft promotes the formation of fog), while
the wind vector anomalies oppose the background flow, indicating more quiescent conditions.
These large-scale patterns are consistent with those that favor consecutive fog events in the UAE,
as discussed in Fonseca et al. (2022), resembling those of the prevailing cluster #1.
**4.  Fog deposition**
In this section, we present fog droplet deposition velocities and liquid water deposition fluxes
calculated based on the EC framework for three fog events measured at the Barakah nuclear site





on January 27, 2022, February 4, 2022, and February 24, 2022. Note that it was not possible to
apply the EC technique to the other fog events prior to these dates because the ultrasonic
anemometer was not yet installed at BNPP.

Figure 7a shows the temporal variation of the gravitational liquid water flux and the settling
velocity computed for the fog case of February 16, 2021. As expected, we observe that these two
quantities are positively correlated when accounting for the sign convention as given by equation
(6). Higher magnitudes of liquid water flux due to the sedimentation process are attributed to larger
water droplets with higher sedimentation velocities. The microphysical mechanisms of
condensation, coalescence, and evaporation impact the number density and size distribution of
water droplets in the fog, which in turn affect the rate of gravitational settling during the duration
of the fog event. Therefore, to investigate the change in gravitational sedimentation rate over time,
we calculated the time-series for the twelve fog events. We calculated settling velocities every 5
minutes, discarding the values associated with visibility greater than 1 km (Fig. 7b). The calculated
gravitational sedimentation velocity for each fog event exhibits a temporal variability which is
most likely determined by the time-variation of the number density of the largest water droplets in
the fog. In addition, we observe that the values of the settling velocities are in the range of 0.1 to
3.5 cm s$^{-1}$. To determine how the distribution of LWC values is related to the gravitational
sedimentation rate, we binned the LWC values over 6 bins of size 100 mg m$^{-3}$ using the composite
data from the different fog events. Thus, for each LWC aggregate, we computed the number of
water droplets and the maximum, minimum, median, mean, and standard deviation values of the
settling velocities, as summarized in Table 4. This calculation confirms that a direct relationship
between LWC in the fog and the gravitational settling rate is not necessarily warranted.

Figures 8a-b show respectively the temporal variation of the fog droplet deposition velocity ($v_d$;
equation (9)) and the associated turbulent kinetic energy flux $\left(\overline{w'^2}\right)$ every 15 min for these three
fog cases. Notice that a positive sign of the fog deposition velocity indicates that more fog droplets
are depositing onto the surface than are being re-entrained into the air. This results in a net gain in
fog deposition for the surface. In contrast, a negative sign of the fog deposition velocity represents
a net loss of fog deposition for the surface.

First, we observe that the fog deposition velocity for the January 27, 2022 case exhibits more
variability with higher magnitude compared to the other two cases. These two latter show
particularly lower values of $v_d$ ranging from 0.1 to 5 cm s$^{-1}$. It is noteworthy that the turbulence
energy associated with the fog of January 27, 2022, is significantly higher compared to the other
two fog events. This indicates that the deposition of fog droplets due to the turbulence mechanism
for January 27, 2022 is larger compared to the other cases. Moreover, it is interesting to note that
$v_d$ for the January 27, 2022 fog event shows a maximum value of almost 40 cm s$^{-1}$, associated
with a peak of the turbulent kinetic energy flux occurring 30 min earlier as shown in Fig 8b. The
averaged values of $v_d$ over the duration of each fog event for the three cases are respectively, 7.8





cm s$^{-1}$ for January 27, 2022, 3.5 cm s$^{-1}$ for February 4, 2022 and 2.1 cm s$^{-1}$ for February 24, 2022.
These calculated values are of the same order of magnitude with the fog deposition velocity values
reported by Tav et al. (2018), in particular with those calculated for the bare soil category. Yet,
they found that the calculated fog deposition velocities for cypress and grass as deposition surfaces
are significantly high, with average values of 16 cm s$^{-1}$ and 40 cm s$^{-1}$ respectively.

Figure 9 shows the 15-min variation of the turbulent liquid water flux, $LWF_{turb}$, the gravitational
settling liquid water flux, $LWF_{grav}$ and the total liquid water flux, $LWF_{total}$, defined by equations
(4), (6) and (7), calculated for 27 January 2022, 04 and 24 February 2022 fog cases. Additionally,
the temporal variation of the total liquid water flux in the fog with respect to the droplet sizes is
also shown. Calculated fluxes are standardized to mg m$^{-2}$ s$^{-1}$. Note that a negative sign of the total
liquid water flux means that there is a net flux of fog water droplets towards the surface, which
results in liquid water deposition on the surface if deposition rate is greater than the evaporation
rate. Conversely, a negative sign of the total liquid water flux indicates a net flow of water droplets
away from the surface, which can result in liquid water removal from the surface.

Interestingly, for all three fog cases, we notice that the magnitude of the $LWF_{total}$ reaches its
maximum for water droplets with larger sizes of 20-45 µm, as indicated by the background maps.
For the fog case of 27 January 2022 (Fig. 9a), we remark that $LWF_{turb}$ is significantly larger
compared to the magnitude of $LWF_{grav}$. This indicates that the fog deposition is mainly dominated
by the turbulent component and that the gravitational sedimentation process plays a lesser role in
this case. The turbulent part is larger than the settling component by a factor of 2.35. This can be
explained by the fact that the atmospheric turbulence during the fog event of 27 Jan. 2022 is well
developed (see Fig. 8b) and, as a result, large fog droplets with greater mass and momentum are
easily entrained by turbulent eddies and are more likely to collide or deposit onto the surface
through the turbulent flow. The cumulative sum of the total liquid water flux converted to
millimeters leads to the value of -0.18 mm as a net gain for the surface. Conversely, we notice that
the fog deposition due to the gravitational sedimentation mechanism prevails over the turbulent
process for both fog events observed on 04 and 24 February 2022. The gravitational settling
component is larger than the turbulent part by a factor of 7 for 04 February 2022, and by a factor
of 1.75 for 24 February 2022 fog case. The time-integrated total liquid water flux over the duration
of each fog event converted to millimeters leads to the values of -0.34 mm and -0.19 mm as net
gain for the surface respectively for 04 and 24 February 2022 fog events. It should be mentioned
that the values of the net flux of deposited water for these three fog events are close to those
reported by Spirig et al. (2021) in their study aimed at investigating the seasonality of fog over the
central arid Namib desert.

In this section, we calculated fog deposition velocities and net fluxes of liquid water deposited at
the surface for three fog events measured at the Barakah site using the EC framework. The values
obtained are of the same order of magnitude as what has been reported in other studies using the
same technique for example as in Spirig et al. (2021), Gultepe and Milbrandt (2007); Westbeld et
al. (2009); Weston et al. (2022), and in Kowalski and Vong (1999). However, it should be noted
that the eddy covariance method may not be reliable in cases where the turbulence regime is not
sufficiently well-developed. Thus, direct measurements of fog water deposition through the setup
of a field experiment at Barakah is vital not only to validate the EC technique but also to obtain
accurate estimates of fog deposition processes at the nuclear site.

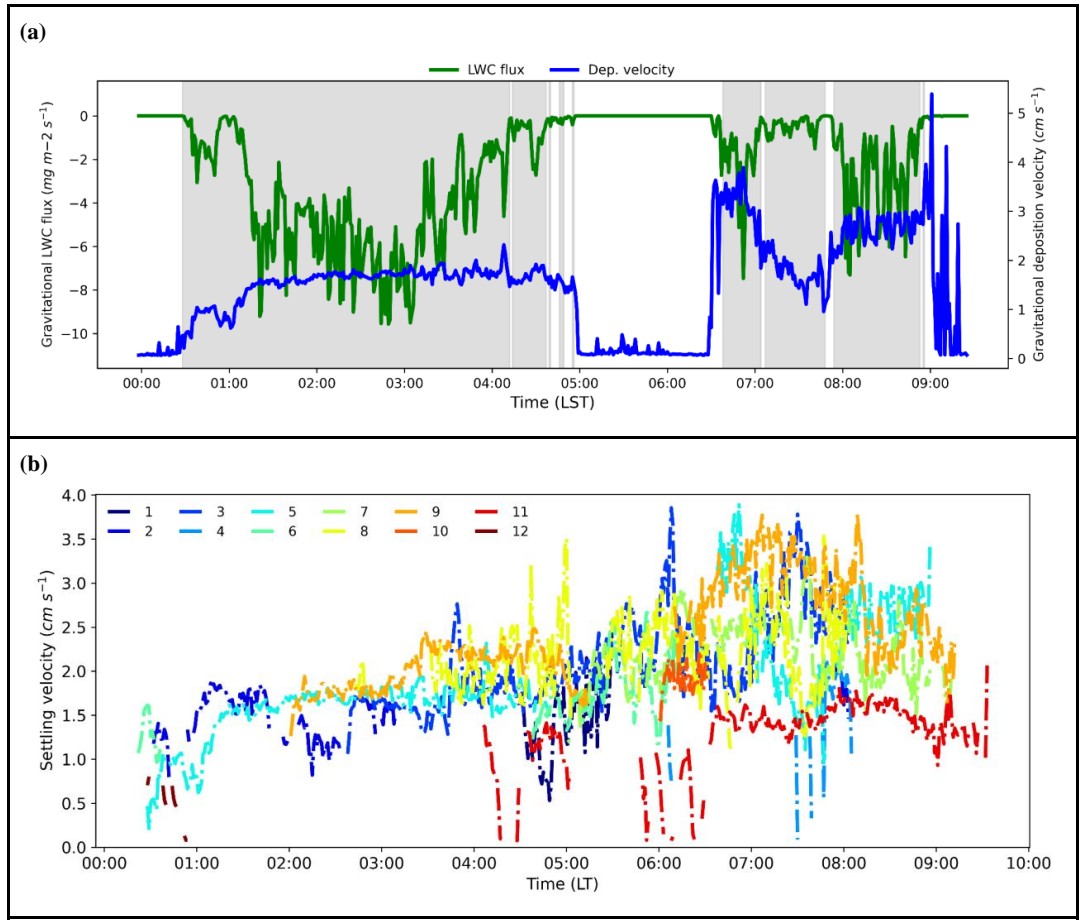

**Figure 7: Temporal variation of the gravitational liquid water flux and settling velocity.** (a) Time-series of the total gravitational LWC flux (green line; $mg\,m^{-2}\,s^{-1}$) and settling velocity (blue line; $cm\,s^{-1}$) for the 16 February 2021 fog event. The gray shading represents regions with visibility lower than 1 km. (b) Time-series of settling velocity ($cm\,s^{-1}$) of the 12 fog cases. The data is averaged every 15 min, and values with visibility above 1 km are discarded.



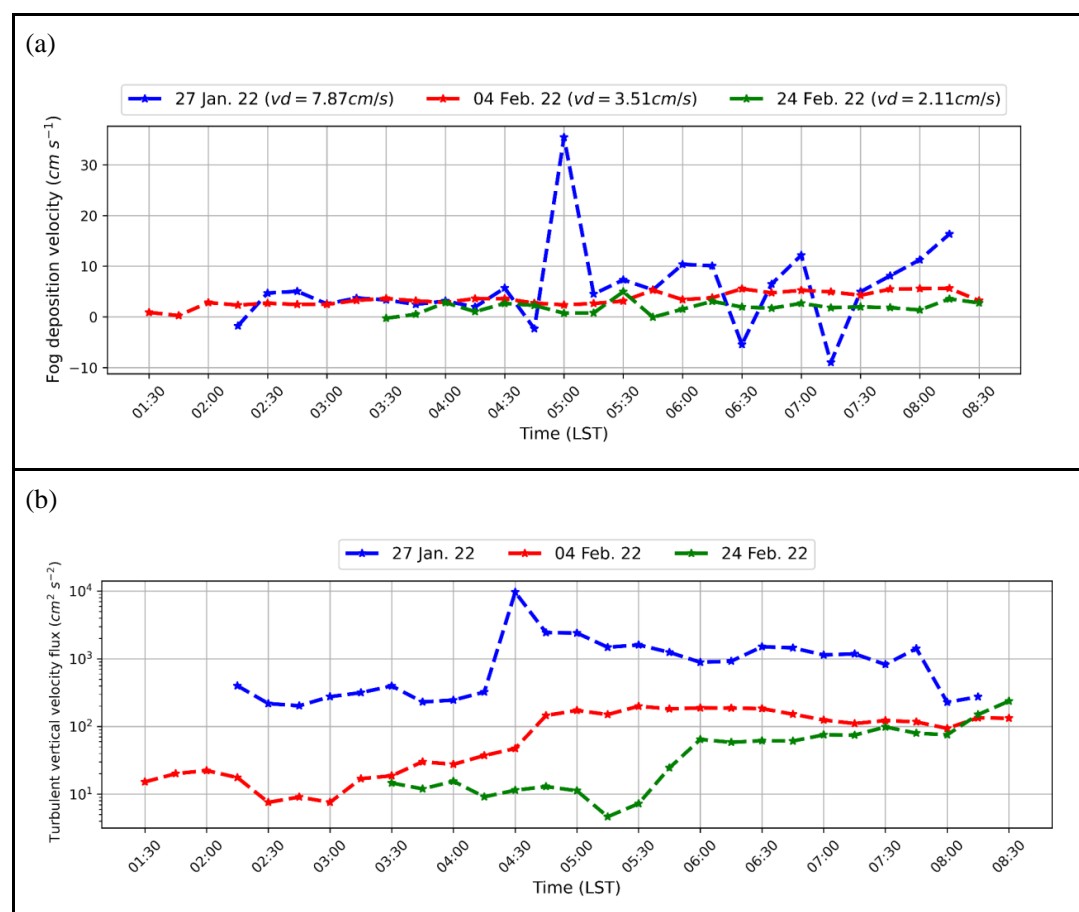

**Figure 8: Deposition and turbulent vertical velocity for 27 January, 04 and 24 February 2022 fog events.** 15 min time-series of the (a) fog deposition velocity (cm s⁻¹) and (b) turbulent vertical velocity flux (cm² s⁻²) for the 27 January, 04 and 24 February 2022. The turbulent vertical velocity flux is given by $\overline{w'^2}$ . The net deposition velocity for each event is given in the caption.






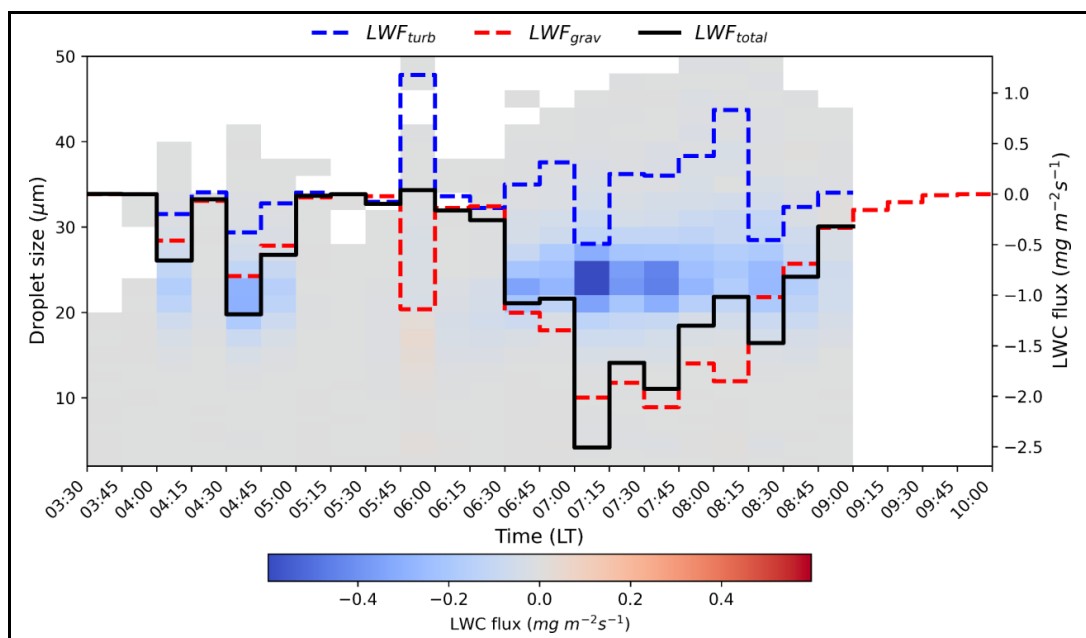

**Figure 9: Liquid water flux for 27 January, 04 and 24 February 2022 fog events.** 15-min liquid water flux $(mg\,m^{-2}s^{-1})$ components calculated for the three fog cases (a) 27 January, (b) 04 and (c) 24 February 2022. The turbulent liquid water flux, $LWF_{trub}$ , is depicted in blue; the gravitational settling liquid water flux, $LWF_{grav,}$ is depicted in red; the total liquid water flux, $LWF_{total,}$is depicted in black. The shading shows the temporal distribution of the total liquid water flux per bin droplet size. Note that EC data for the 24 February 2022 fog event was not available for the entire duration of the fog.



**Table 4:** Distribution of deposition settling velocities for different LWC bins.

| LWC (mg m⁻³) | min (cm s⁻¹) | max (cm s⁻¹) | mean (cm s⁻¹) | std (cm s⁻¹) | count |
|---|---|---|---|---|---|
| 000-100 | 0.08 | 3.77 | 1.95 | 0.65 | 1586 |
| 100-200 | 0.79 | 3.89 | 2.0 | 0.54 | 362 |
| 200-300 | 0.84 | 3.47 | 1.91 | 0.51 | 128 |
| 300-400 | 1.48 | 3.86 | 1.95 | 0.55 | 69 |
| 400-500 | 1.59 | 2.59 | 1.74 | 0.16 | 34 |
| 500-600 | 1.59 | 1.9 | 1.73 | 0.1 | 10 |




### 5.  Impact of fog on dispersion and deposition of radionuclides

The goal of this section is to investigate the impact of fog on ground deposition of radioactive materials in case of an accidental radioactive release at Barakah Nuclear Power Plant (BNPP). To this end, we first use the Weather Research and Forecasting Model (WRF) to carry out a 3D non-hydrostatic numerical simulation for a two-day period covering the fog event of 16th Feb. 2021. Then, the generated WRF-forcing data are used to drive a state-of-the-art Atmospheric Transport and Dispersion Model (ATDM) to simulate atmospheric radionuclide dispersion from a hypothetical accidental radioactive release at BNPP.

Fog simulation is sensitive to the choice of the PBL parameterization scheme. Therefore, we performed five 48-hour WRF simulations, using three local PBL schemes, MYJ, MYNN2.5 and MYNN3.0, and two non-local PBL schemes. The model is initialized at 00 00 UTC on 15th Feb. 2021. The first twelve hours are discarded as spin-up time and the model output data were recorded every 15 minutes.  The innermost nested domain, which represents the study area, is illustrated in Fig. 1b. These five model experiments would help evaluate the overall model's performance for simulating the specific fog event observed on 16 Feb. 2021. Selecting the right PBL scheme is essential to ensure accurate fog simulation results but at the same time it is a very difficult task in modeling. However, it should be noted that the sensitivity test carried out in this study may provide insights into the sensitivity of the model to various parameters related to the planetary boundary layer. Having said that, it does not aim to provide a comprehensive assessment of the model's performance for simulating fog over this hyper-arid region.

In the context of fog, the LWC is an important parameter for understanding the formation, maintenance, and dissipation of fog. Fig. 10 shows a qualitative comparison between simulated LWC by WRF and SEVIRI Fog RGB map for the fog event recorded on 16 February 2021 at 00:30 UTC. Figs. 10b and 10c depict the spatial patterns of the simulated LWC obtained by averaging over the three lowest model levels (0-60m) for the nonlocal, ACM2, and the local, MYJ PBL schemes, respectively. The SEVIRI RGB map (Fig. 10a) shows a band of fog along the entire coastal regions of the UAE with a large fog patch near BNPP. The simulated LWC highlights that both the nonlocal and the local PBL schemes are able to provide reasonable spatial patterns in comparison with the SEVIRI RGB map. Besides, the nonlocal PBL scheme shows higher values of LWC particularly near to the nuclear site and wider fog extent than the local PBL scheme. This finding is in perfect agreement with Chen et al. (2020) results.

Figure 11a presents the modeled LWC from the first three model levels (0-60m) with the different PBL schemes, along with the observed LWC for the fog case on 16 February, 2021. Figure 11b, shows the diagnosed model visibility for the different PBL schemes using a power law relationship



710 that links the visibility to LWC as follows: $Vis = \alpha \times LWC^{\beta}$. This relationship is fitted using
711 visibility and LWC observations for the twelve fog cases. The regression coefficients $\alpha, \beta$ have the
712 values 0.94641 and -0.31572, respectively (Fig. A5). The LWC and visibility observations indicate
713 that the fog started at 00:30 (LST). With the lower nighttime temperatures, the fog lasted for 4 hr
714 and 15 min until 4:45 am, burning off one hour before sunrise. It is noteworthy that even though
715 most of the fog had swiftly dissipated just after sunrise, there were still some fog patches that
716 remained until 9 am. They may have been transported by wind towards the site, where it was
717 captured by the fog droplet monitor and the visibility meter.

719 As shown in Fig. 11, and for this specific fog event, we notice that the different PBL schemes used
720 in the model simulation have varying levels of accuracy in simulating the correct onset and
721 dissipation times of fog, especially between local and nonlocal schemes. Furthermore, we observe
722 that the simulated LWC in all the numerical experiments has higher values than the observations.
723 Especially for nonlocal schemes (ACM2, YSU), where the modeled LWC values are almost twice
724 as high. This result was also pointed out by Chen et al. (2020) and Ghude et al. (2023). Except for
725 the nonlocal ACM2 scheme, all other PBL schemes were able to simulate the fog onset time almost
726 exactly. However, they failed to capture the correct fog dissipation time as they showed earlier
727 dissipation times than those observed (with an average of 2 hours). In contrast, the nonlocal ACM2
728 scheme simulated the fog onset time 1 h late and appears to capture the fog dissipation time
729 reasonably well. Remarkably, we note that the local MYJ scheme succeeded in capturing the
730 remnant of the fog while all other PBL schemes have failed.

732 Here, we use meteorological output data from the WRF-MYJ model experiment to drive
733 FLEXPART in forward mode to simulate the transport and deposition of radionuclides at high-
734 spatial resolution (1 km) due to a hypothetical accidental radioactive release occurring at BNPP.
735 A radioactive release of $^{137}$Cs as aerosol isotope with an emission rate of $10^{14}$ Bq h$^{-1}$ over a 28-
736 hour duration is considered. $^{137}$Cs has a large radiological soil contamination impact due to its long
737 half-life (~ 30 years).

739 In order to highlight the impact of the fog deposition process on the ground deposition of
740 radionuclides, we perform two dispersion simulations for the case of the fog of 16 February, 2021.
741 The first simulation is run with the default parameters of the dispersion model, while the second
742 is run with the modification made to the gravitational sedimentation scheme as described earlier.
743 In both, FLEXPART is integrated over 48 hours from 00:00 UTC on 15 February, 2021, and the
744 radioactive release is assumed to start 20 hours after initialization. The dispersion model output
745 data is recorded every 15 minutes. Figures 12a-b show the spatial distribution of the total
746 deposition of $^{137}$Cs to the ground (sum of dry and wet deposition) integrated from 08:15 to 08:30
747 LT on 16 February, 2021, respectively from the baseline simulation and that carried out under
748 foggy conditions. As expected, we observe a noticeable downwind increase in ground deposition
749 caused by the fog deposition process, which will subsequently lead to higher levels of soil





contamination. To gain more insight into how the fog occurrence impacts the temporal variation
of the ground deposition, we computed the time-variation of the contaminated land areas of $^{137}$Cs
for a level value of 200 Bq m$^{-2}$. The contaminated area is calculated by counting the number of
land grid cells in the model domain for which combined values of dry and wet deposition are
greater than 200 Bq m$^{-2}$. Each model grid cell has an area of 1 km$^2$. Figure 12c shows the temporal
variation of the contaminated areas for both the reference dispersion simulation and that with the
effect of fog deposition included. We notice that the contaminated land areas of the two dispersion
simulations show similar temporal variation patterns, and a gradual increase to reach their first
peaks around 10:00 LT, a few hours after the accidental release. Besides, the contaminated land
areas calculated under foggy conditions show higher values than those of the baseline simulation,
particularly from 00:00 to 12:00 LT where the fog event was measured at BNPP. It should be noted
that after the complete dissipation of the fog around 9:00 a.m. LT, the values of the contaminated
land region considering the fog deposition begin to gradually decrease to have values similar to
those of the reference simulation. In addition, to quantify the relative contribution of the fog
deposition process to the total ground deposition (dry and wet deposition combined) of $^{137}$Cs, we
first integrated in time the two dispersion simulations over 28 hours after the start of the radioactive
release. Then, we calculated the ratio of the deposition field under foggy conditions to that of the
simulation of reference. Figure 12d shows the spatial distribution of this ratio given in percentage.
Remarkably, we observe that the fog deposition contributed to the total ground deposition by 30-
40%, and regions with marked increases in ground deposition are located near to the source. In
addition, it should be noted that the effect of the fog deposition is relatively small near the edges
of the radioactive plume where strong gradients exist.
In this section we have performed a simple modeling simulation experiment to highlight the
potential impact of fog on radionuclide ground deposition. It has been demonstrated in this test
case that the fog deposition enhances the deposition of $^{137}$Cs radionuclide on the ground by acting
as a scavenging agent. The actual impact of fog on radionuclide deposition can vary widely
depending on the specific situation. Moreover, the solubility and chemical form of the
radionuclide-labeled particles can significantly influence this impact. For example, in cases where
the radionuclides are highly volatile or have a short half-life, fog may not have a significant effect
on their deposition. In other cases, the wind direction and speed can also play a role, as they can
affect how the fog and radionuclides are dispersed and transported. Yet, further modeling efforts
are needed to incorporate an accurate fog deposition scheme into the WRF model or into the
dispersion atmospheric model fitted to the BNPP region.



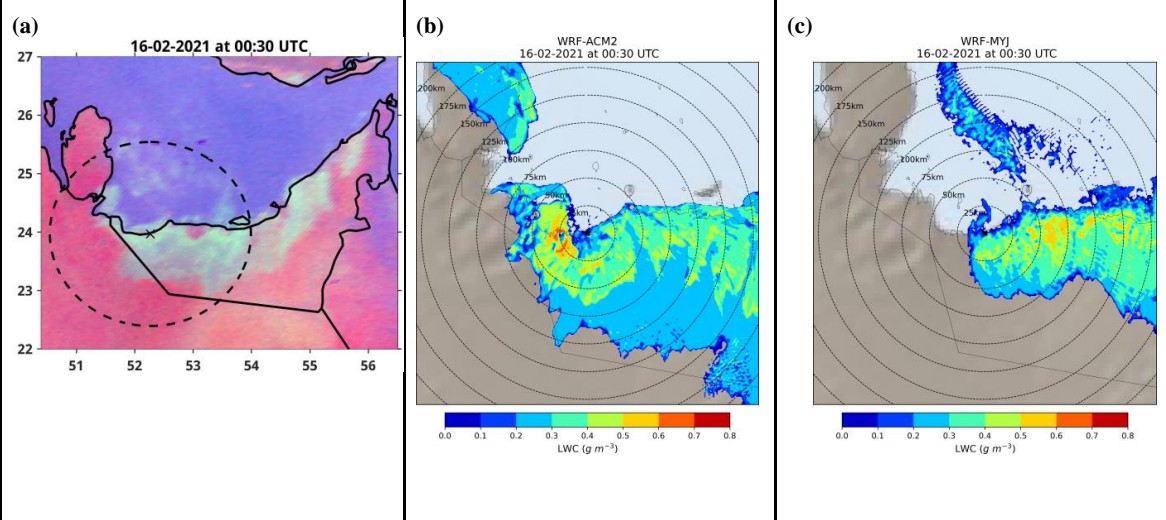

**Figure 10.** Comparison between WRF simulation and SEVIRI Fog RGB on 16 February 2021 at 00:30 UTC. (a) SEVIRI Fog RGB map. The dashed circle has a radius of 200 km and is centered at Barakah. (b) and (c) show the WRF simulated spatial pattern of LWC averaged over the three lowest model levels (0-60 m) using respectively the nonlocal (ACM2) and local (MYJ) PBL schemes.




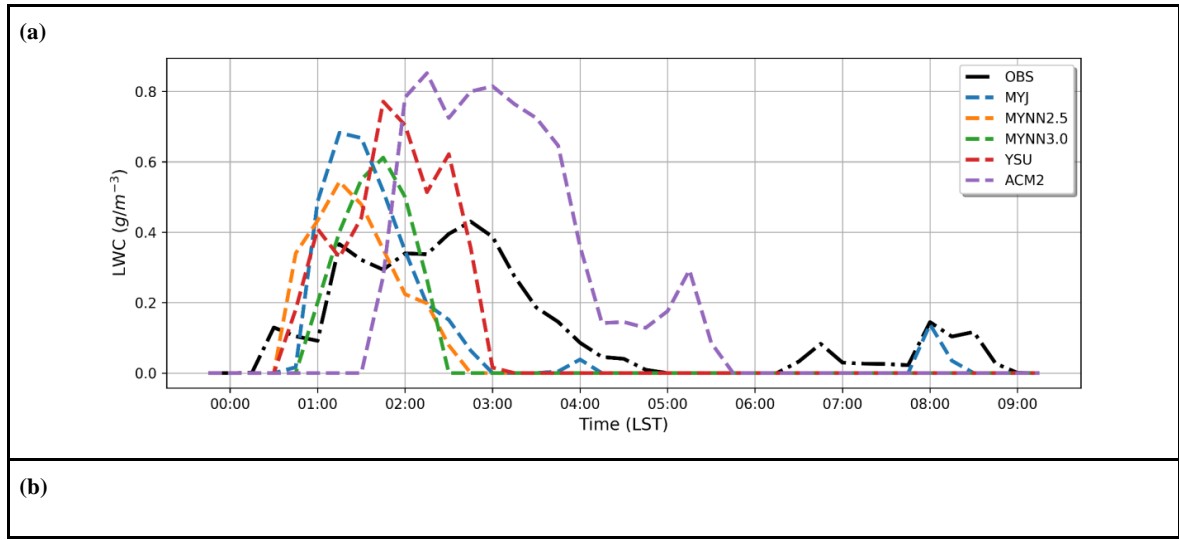





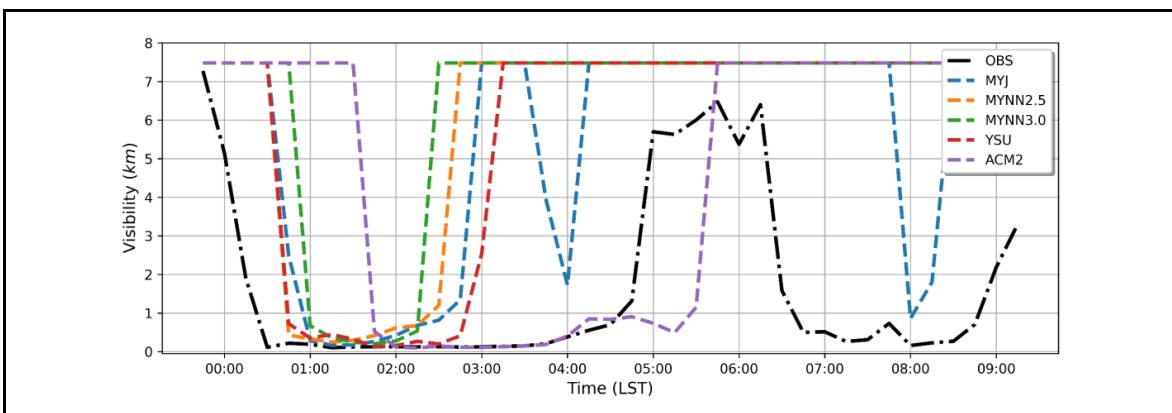

**Figure 11.** Comparison between WRF simulations and FM-120 observations for 16 February 2021. (a) Simulated LWC (g m$^{-3}$) using the three first model levels (0-60m) for the MYJ (blue), MYNN2.5 (orange), MYNN3.0 (green), YSU (red) and ACM2 (purple) PBL schemes and that observed (black) in 15 min intervals. (b) is as (a) but showing the model-diagnosed and observed visibility (km). The model-diagnosed visibility is obtained by fitting a power law relationship that links the visibility to the LWC values using observations for the twelve fog cases: $Vis = \alpha \times LWC^{\beta}$ with $\alpha = 0.94641$ and $\beta = -0.31572$.
















**Figure 12:** Quantification of the effect of fog deposition on the total radionuclide's deposition (combined dry and wet). (a) and (b) spatial distribution of the total surface deposition of $^{137}$Cs integrated from 08:15 to 08:30 LT on 16 February 2021, respectively for the baseline simulation and that carried out under foggy conditions. (c) Temporal variation of the contaminated land area for the level 200 Bq/m$^2$. The contaminated area is calculated by counting the number of land grid cells in the model domain for which combined values of dry and wet deposition are greater than 200 Bq m$^{-2}$. Each model grid cell has an area of 1 km$^2$. (d) Fractional contribution (%) of fog deposition to the total surface deposition of $^{137}$Cs.





## 6. Conclusions

The main objective of this paper was to characterize the deposition of fog droplets around the Barakah nuclear site and to perform a modeling simulation experiment to explore the impact of fog deposition on the ground deposition of radionuclides in the event of a radiological emergency at the site leading to the release of radioactive materials into the air.

State-of-the-art in-situ instruments including a ground-based radiometer, fog monitor sampler, ultrasonic anemometer, and visibility meter have been deployed around the site to probe fog events during the winter seasons of 2021 and 2022, and twelve cases of fog were recorded. The microphysical properties for each fog event were inferred by analyzing data of horizontal visibility, liquid water content (LWC), number concentrations ($N_c$), and median volume diameter data (MVD). Analysis of the size distribution of the mean number concentration for each fog event revealed that all observed fogs exhibit the same bimodal distribution shape, with modes at 4.5 and 23.2 $\mu m$, indicating the coexistence of two distinct populations of droplets in the fog, with a high proportion of small droplets compared to the number of large droplets. The size distribution of the mean LWC showed also a bimodal distribution with a broadening peak at 5.5 $\mu m$ and a pronounced peak at 25 $\mu m$. This confirmed that the greatest contribution to the LWC originates mainly from large droplets, between 15 and 35 $\mu m$, despite their lower number concentration. The time-varying number and mass densities revealed that at the onset stage, the smaller to medium-sized droplets (1-10 $\mu m$) formed by condensation of water vapor contribute more to the LWC and account for up to 85% of the total number of droplets. However, the large droplets (10-50 $\mu m$) formed mainly by the collision-coalescence process at the mature stage, representing less than 20% of the total number of droplets. They account for up to 90% of the total condensed water. Moreover, the MVD ranges from 20 to 26 $\mu m$ in almost all cases, suggesting that the radiation fogs observed in this region tend to form larger droplets.

The microphysical mechanisms of condensation, coalescence, and evaporation impact the number density and size distribution of water droplets in the fog, which in turn affect the rate of fog droplet deposition on the surface. This study is the first application of the Eddy Covariance (EC) framework to measure the fog droplet deposition velocity to a hyper arid coastal site. The EC technique was applied for three fog events recorded at the Barakah site on January 27, 2022, February 4, 2022 and February 24, 2022. The LWC flux in the fog is calculated as the sum of the turbulent and gravitational settling components. As expected, higher amplitudes of liquid water flux are found to be attributed more to larger-size droplets. For the fog case of January 27, 2022, the fog deposition is found to be primarily controlled by turbulence, with a net gain deposited water for the surface of about -0.18 mm, and an average deposition velocity of 7.9 cm s$^{-1}$. Conversely, the fog deposition was found to be largely attributed to the gravitational sedimentation process for the fog events of February 4, 2022, and February 24, 2022, with net gains for the surface of about -0.34 mm, and -0.19 mm, and average deposition velocities of about 3.5cm s$^{-1}$, and 2.1 cm s$^{-1}$ respectively.

The calculated net fluxes of water deposited on the surface are found to be of the same order of magnitude as what has been reported in some studies using the same technique (e.g., Kowalski and



Vong, 1999; Spirig et al., 2021, Gultepe and Milbrandt, 2007; Westbeld et al., 2009 and Weston et al., 2022). Moreover, the typical values of the average fog deposition rate calculated for these three fog events are of a similar magnitude to the values reported by Tav et al., (2018) for the bare surface type. However, it should be noted that the EC method may not be reliable in cases where the turbulence regime is not sufficiently well-developed.

In order to assess the impact of fog deposition on ground deposition of radionuclides released into the air in case of an accidental radioactive release occurring at BNPP, a modeling simulation experiment was carried out. The ratio of the time-integrated ground deposition of $^{137}$Cs under foggy conditions to that of the baseline simulation, showed that the fog deposition contributed to the total ground deposition of $^{137}$Cs by 30 to 40%. This result demonstrated that incorporating the fog deposition process in dispersion modeling as an additional scavenging mechanism is vital under foggy conditions. This is consistent with previous findings; in France for instance and based on model simulations, it has been found that the presence of a fog during a nuclear accident could lead to an almost twice higher surface contamination density of $^{137}$Cs than without fog and with only dry deposition (e.g., Masson et al., 2015).

Our findings are useful for future work on fog deposition and field campaigns in this semi-arid region. The study will be extended by applying the Eddy Covariance technique to estimate the fog-droplet deposition rates to additional fog events. It would be enlightening to plan a field experiment to acquire direct measurements of fog deposition at the site by a sensitive weighing method, and collection of fog water for further analysis. This will help not only to validate the EC technique but also to better estimate the fog deposition parameters. Also, this would help in evaluating the performance of the fog deposition scheme in the WRF model. Additionally, further modeling efforts are required to efficiently account for the fog deposition as an additional removal process in dispersion modeling.

## Acknowledgment

We would like to thank Khalifa University's high-performance computing and research computing facilities for their support of this research work. This research work was supported by the Federal Authority for Nuclear Regulation (FANR) through the research project Modeling of Radionuclides Dispersion in the UAE Environment (MORAD). Nawah and SHAMS power companies are both acknowledged for facilitating the deployment of instruments and conducting the field campaign in the UAE desert.

## Conflict of Interest

The authors declare they have no conflict of interest.

## Appendix




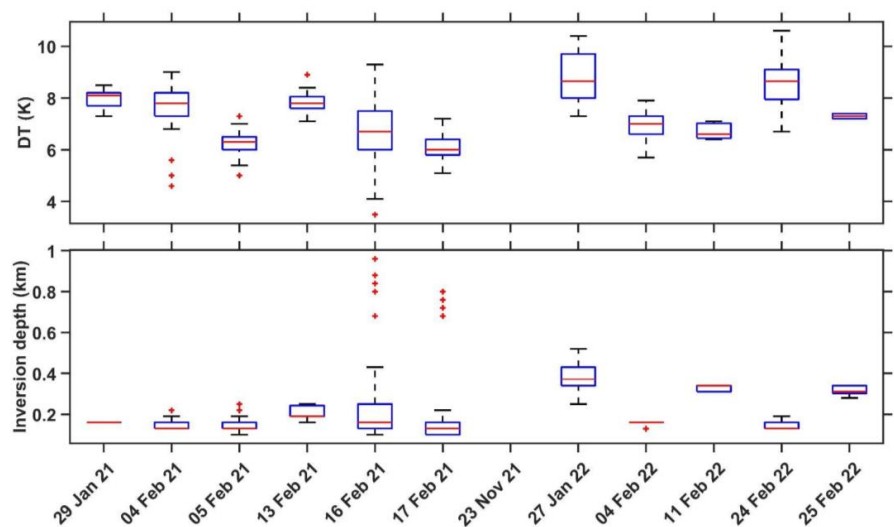

**Figure A1: Temperature inversion during 12 fog events.** Surface-based temperature inversion strength (top; K) and depth of the inversion layer (bottom; km) for the 12 fog events.




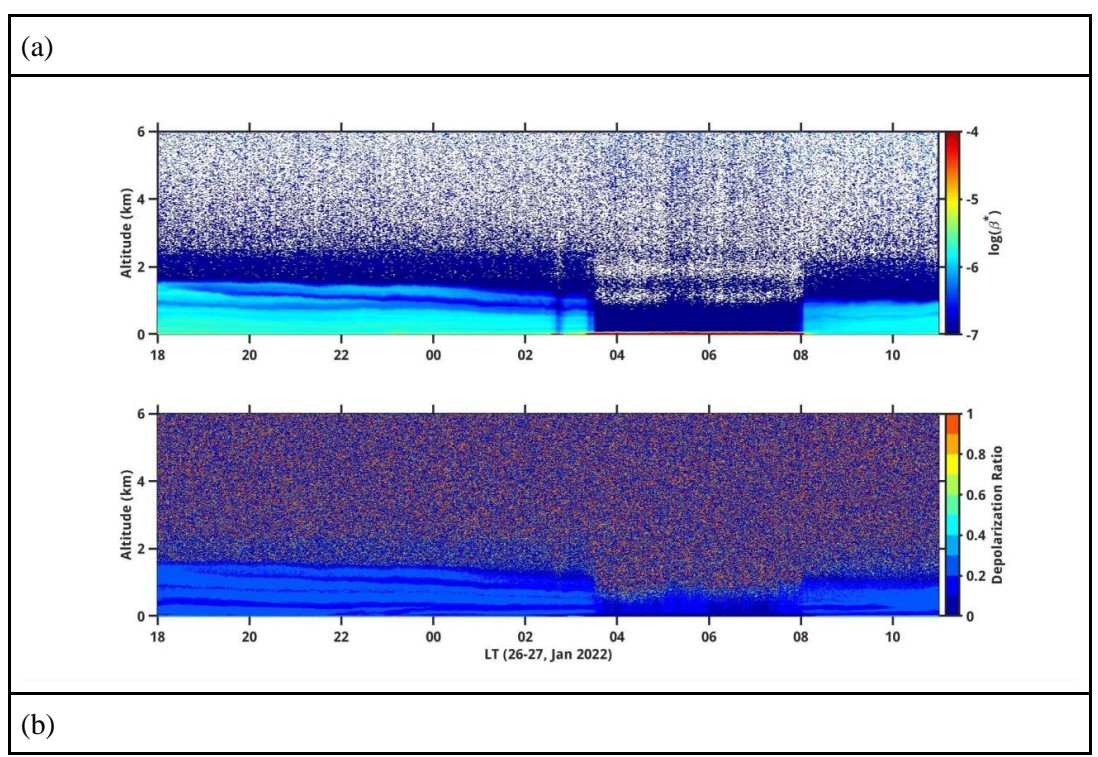





**Figure A2: Attenuated backscatter profiles from Ceilometer Lidar**. The profiles are shown for the (a) 27 January, (b) 04 and (c) 24 February 2022 fog events. The top plot shows the backscatter coefficient, log ($\beta^*$), while the bottom plot shows the depolarization ratio (0-1).





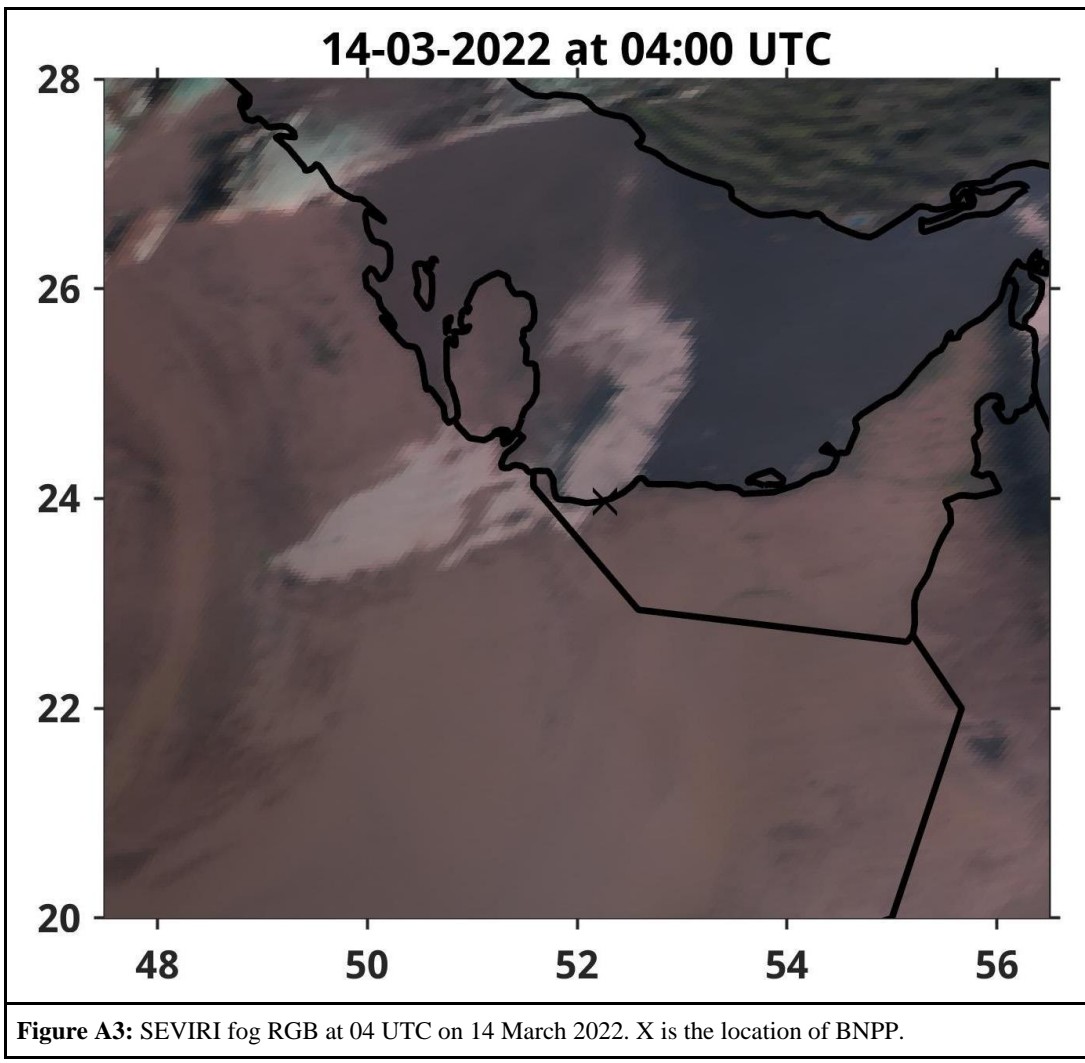

**Figure A3:** SEVIRI fog RGB at 04 UTC on 14 March 2022. X is the location of BNPP.


(a) ERA-5 full fields



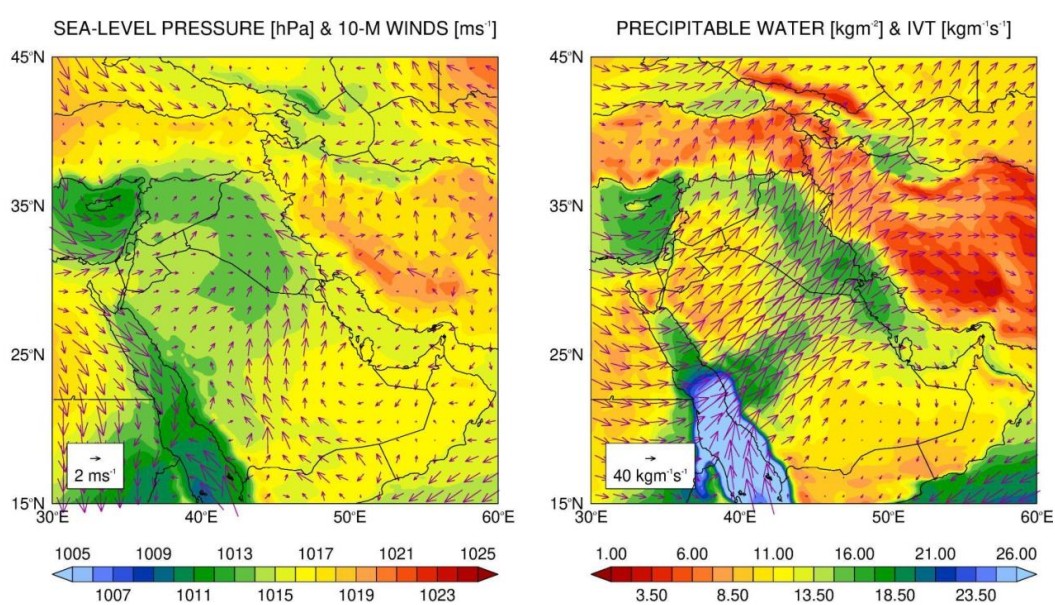

(b) ERA-5 anomalies

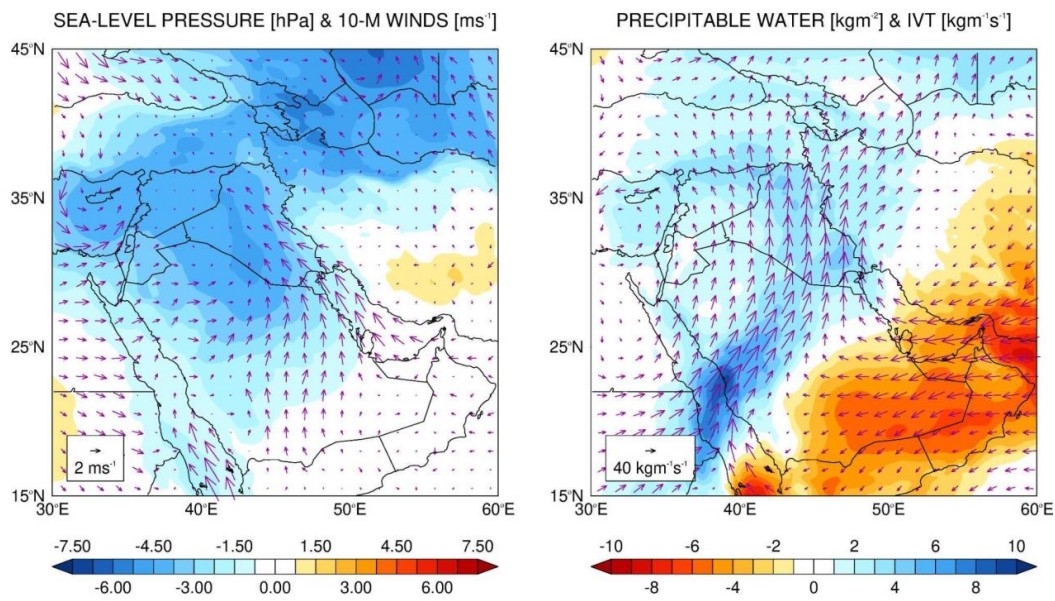

**Figure A4: ERA-5 fields averaged over all 12 fog events.** (a) Sea-level pressure (shading; hPa) and 10-m wind vectors (arrows; m s$^{-1}$) (left) and precipitable water (shading; kg m$^{-2}$) and integrated vapour transport (arrows; kg m$^{-1}$ s$^{-1}$) (right) averaged over 00-04 UTC of the 12 fog days listed in Table S1. (b) is as (a) but showing anomalies with respect to ERA-5's 1979-2021 monthly climatology at the respective hour of day.




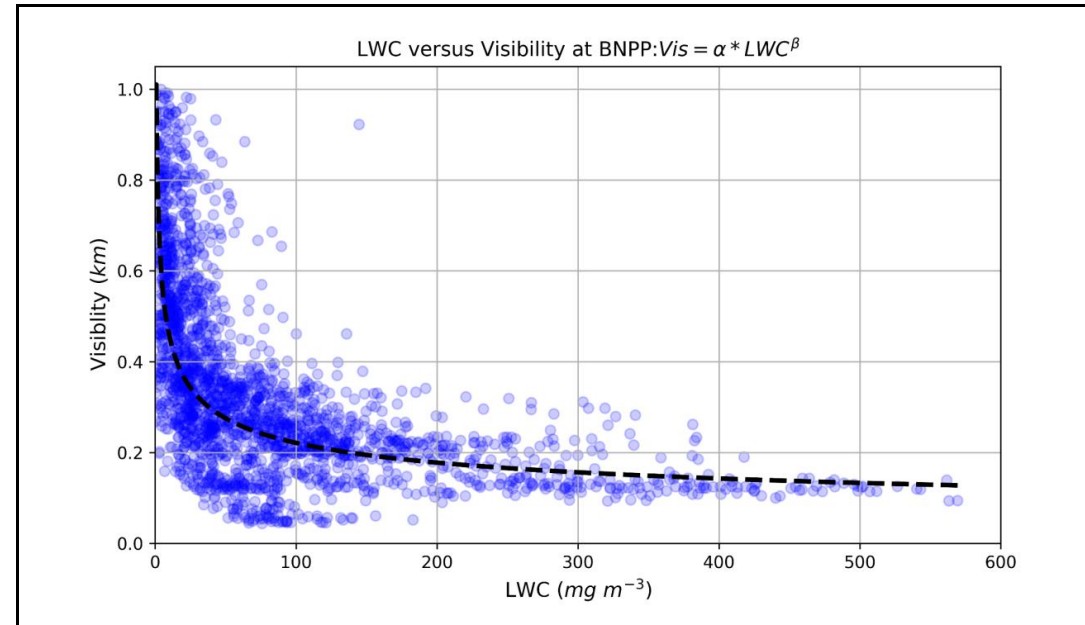

**Figure A5: Scatter plot of LWC versus visibility at BNPP.** The black dashed line depicts the fitted power law relationship that links visibility to the LWC values using observation data of the twelve fog events: $Vis = \alpha \times LWC^{\beta}$ with $\alpha = 0.94641$ and $\beta = -0.31572$. Each individual observation is shown by a filled blue circle.

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
