# Peer review of "Microphysics of radiation fog and estimation of fog deposition velocity for atmospheric dispersion applications"

_EGUsphere, 2023_

## Referee Comment (RC2)

Reviewer comments to authors of "Microphysics of radiation fog and estimation of fog deposition velocity for atmospheric dispersion applications"

In this paper, the authors have undertaken the task of characterizing radiation fog in around Barakah nuclear power plant in the United Arab Emirates. They have observed twelve fog events and conducted an analysis of microphysical and dynamical properties, and thermodynamics. Furthermore, they simulated a specific fog event (15 Feb. 2021) using various PBL schemes. The authors also delved into analyzing a particle dispersion model to comprehend the impact of fog deposition. Overall, this paper encompasses a wide range of analyses lack a thorough and cohesive understanding, making it challenging to arrive at a solid conclusion.

One notable concern is that fog events differ in each section and figure. For instance, section 3.1 focuses on the fog event of Feb 25, 2021 (Figure 5), while section 3.2 deals with fog events on Jan 27, Feb 4, and Feb 24, 2021 (Figures 6c and 6d). Similarly, section 4 concentrates on the Feb 16, 2021 fog event (Figure 7), whereas section 4 involves Jan 27, Feb 4, and Feb 24 (Figures 8 and 9). Section 5 centers around Feb 15. The reviewer is perplexed as to why different fog events are chosen for each section, leading to confusion regarding the author's primary focus. Furthermore, the connection between the WRF simulations and the preceding analyses (such as fog microphysical characteristics and the thermodynamic environment) is not well established. The instruments used to observe the fog events also appear to yield differing results.

In light of these issues, the reviewer strongly recommends that the authors narrow down their research topic and engage in more in-depth analyses. Addressing these concerns is crucial for enhancing the manuscript's coherence and overall quality. Given the current state of the manuscript, it is the opinion of the reviewer that it may not be suitable for publication in ACP.

Minor comments:
1. Line 76: Remove the semicolon (;)
2. Line 90-91: I am not clear what the sentence means. Also, it might need to consider a referent to support author's argument.
3. Line 101: If the author would like to emphasize this (first observation-derived fog deposition velocity for the UAE), it would be great if the author presents a distinct feature about the first study by comparing the previous studies with different regions.
4. Figure 1: It could be very helpful the author can mark the observational sites on the map, because the instruments installed the different sites. Also, please change the color for the ocean, because it looks land since the lowest elevation over land also shaded blue color.
5. Table 1: It could be very helpful if the author can add the measurement bias at each instrument.
6. Table 1: There were two analysis period in one box. Please modify the box correctly. The period was not same among the instruments. How the author analyzes the data? Does them use the same period of case?
7. Line 221: Why the author uses the WSM3 rather than WSM6 as microphysics scheme? The WSM6 provides the 6 class hydrometeor types including ice, snow, and graupel.

8. Line 289-290: Form what?
9. Figure 2: at Barakkah? Does this mean at BNPP?
10. Line 509-511: Need result/figure or reference to support the argument.
11. Figure 6a,b: Hard to see the lines
12. Line 623-624: How can see this result? The reviewer can not find the result correctly in Figure 9a. The background map showed wide ranges. Please explain more about this.
13. Line 677-678: What the author refers to? FLEXPART?
14. Figure 10: Why the author shows only two simulations (ACM2 and MYJ)? The simulations were performed five (Table 2), so please add more figures for all simulations.

---

## Author Comment (AC1)

**Response to Reviewer #1 comments on Manuscript egusphere-2023-956: Microphysics of radiation fog and estimation of fog deposition velocity for atmospheric dispersion applications**

This manuscript is a detailed description of studies of fog physics in the Abu Dhabi region in the winter seasons 2021 and 2022. Twelve events were analyzed, while not all instrumentation was operative during all 12 events. Nevertheless, the data sets bear a lot of information. The manuscript is long, providing much information about fog physics and meteorological conditions. Much of the description provided does not seem focused in terms of the scientific goal of the manuscript.

We appreciate the reviewer's feedback on our work, and thank him/her for his/her several comments/suggestions that help to further improve the quality of our study. We have shortened the paper and sharpened our message, placing greater emphasis on the main scientific goals of the study. Below we address each separately, highlighting in the text where changes, if any, were made.

This reviewer has 2 major concern:

(1) How was the ultrasonic anemometer located with respect to the FM120 fog spectrometer? The photographs in Figure 1 suggest that these 2 instruments were not co-located. That would lead to the conclusions that all eddy covariance (EC) data with respect to turbulent LWC and fog droplet fluxes were void. Authors need to either present convincing arguments that the positions of the 2 instruments were close enough to each other (please provide exact details) for EC application, or to delete all information and data from the manuscript that refers to fog EC.

Reply: We appreciate your meticulous observation regarding the co-location of the 3D ultrasonic anemometer and the FM120 fog spectrometer, and acknowledge the potential implications for the eddy covariance (EC) data interpretation. The 3D ultrasonic anemometer is strategically installed at a height of 4 m on a 10-m meteorological tower. The positioning of this anemometer is approximately 6 km upwind from the FM120 fog monitor's location. We concur with your concerns regarding the inherent uncertainties tied to the estimation of the liquid water turbulent flux due to the spatial separation between these two instruments.

The logistical constraint of our study is that the anemometer is situated at a remote location where a power source, essential for operating the fog monitor, is unavailable. However, we would like to highlight that the fog events under investigation exhibited spatial homogeneity. In other words, variations in droplet size, density, and turbulence characteristics are minimal between the two locations. This homogeneity feature is a valid hypothesis in this study, in particular because the

area where the two instruments are installed is a hyper-arid environment with flat terrain and homogeneous land cover (bare earth), which most likely will lead to low spatial changes in the fog deposition rate. This is exemplified in the SEVIRI fog Red-Green-Blue images for the three events presented in Figure 8, which we have referenced below (Figure R1). As the reviewer can see, the fog patches encompass almost whole UAE and the two locations (6 km apart from each other's) are completely under the same fog cloud.

| 01 UTC, 27 Jan, 2022 | 01 UTC, 04 Feb, 2022 | 04 UTC, 24 Feb, 2022 |
|---|---|---|

[Figure]

**Figure R1.** SEVIRI Fog Red-Green-Blue (RGB) plots for the 27 January, 04 and 24 February 2022 events, shown at 01 UTC, 01 UTC and 04 UTC, respectively.

Furthermore, we have compared the meteorological parameters at both sites. The correlation coefficients for air temperature and relative humidity between the two locations for the three fog events in 2022 are 0.98 and 0.96, respectively. This provides further evidence of the similar meteorological conditions at both sites during the fog events.

In light of the reviewer's feedback, we have re-evaluated our presentation of the EC data in the manuscript, ensuring that we clearly communicate the limitations and provide an adequate justification for our approach (Lines: 188 – 208, section 2.2.4).

(2) Please provide evidence that a data collection rate of 1 Hz (line 145) is sufficient to compute reasonable fog LWC fluxes.

Reply: Thank you for highlighting the need for clarity regarding the data collection rate. We regret the oversight in our initial submission. For precision, our 3D ultrasonic anemometer records measurements at a frequency of 10 Hz, while the fog monitor operates at 1 Hz. We begin by determining the perturbations in the vertical velocity component using the 10 Hz data from the anemometer. It is essential to note that the fog monitor's operational capability is capped at a 1 Hz sampling time. The sampling times for all instruments used in this study have now been added to Table 1. Therefore, for the purpose of calculating fog deposition flux, we average the vertical velocity perturbations over a 1-second interval. These averaged perturbations (w') are subsequently utilized to estimate the fog deposition flux.

Turbulence in the atmosphere is characterized in terms of its energy distribution across different scales. The Kolmogorov spectrum for the inertial subrange describes this turbulent energy distribution across frequency bands. Within this subrange, turbulent energy predominantly transfers from larger to smaller scales without major production or consumption. The energy spectrum can be expressed as $E(f) \propto f^{-\frac{5}{3}}$, $f$ denoting the frequency (Stull, 1988). Our 10 Hz sampling captures most of the turbulence within this subrange. Averaging these to a 1-second interval acts as a low-pass filter, retaining the larger, more energetic eddies relevant for fog deposition flux. These large eddies are fundamental to the vertical transport in the atmosphere, while higher frequency, smaller eddies contribute less to this transport (Stull, 1988). When monitoring vertical fluxes, the focus is often on the time-averaged fluxes, enabling the use of considerably lower sampling rates. For illustration, (Bosveld and Bouten, 2001) utilized a 1 Hz sampling rate for eddy-covariance measurements made 30 m above an 18 m coniferous forest. Hence, our methodology is aligned with the scales most relevant to fog deposition flux and the principles of atmospheric turbulent spectra. It should be noted that the sampling rate also depends on the complexity of the environment where the eddy-covariance system is operating. In our case, both the anemometer and the fog monitor are deployed in hyper-arid, flat terrain characterized by bare land. These additional clarifications have been included in the revised version of the manuscript (Line: 466 – 488, section 2.6). We trust this provides clarity on our methodology, and we appreciate your attentive feedback.

**Further comments:**

1.  In the abstract, please explain the acronyms MYJ PBL scheme and FLEXPART upon their first mention

    Reply: We apologize for the oversight. In the abstract, "MYJ PBL scheme" stands for the Mellor-Yamada-Janjic Planetary Boundary Layer scheme, while "FLEXPART" denotes the FLEXible PARTicle dispersion model. We have now expanded these acronyms both in the abstract and within the main text (Line: 24-25).

2.  line 27: The precision of the number 23.16 is too high.

    Reply: Thank you for pointing that out. We have adjusted the value to a more appropriate precision level in the revised manuscript for better clarity and readability (Line: 28).

3.  line 32: The precision of the numbers 2.11 and 7.87 is too high. Please use only 2 significant digits. In this case: 2.1 and 7.9.

    Reply: Thank you for your suggestion. We have adjusted the values of 2.11 and 7.87 to 2.1 and 7.9, respectively, in the revised manuscript (Lines: 33).

4. line 163, Table 1: Replace Luft by Lufft

   Reply: We apologize for the oversight. The reference to "Luft" in text and Table 1 has been corrected to "Lufft" in the revised manuscript (Line: 176). Thank you for bringing this to our attention.

5. Line 214: Re-type forty-five

   Reply: Thank you for pointing it out. We have corrected the text in line xx, replacing the word form "forty-five" with its numeral representation "45" in the revised manuscript (Line: 253). We appreciate your diligence.

6. Lines 282 – 285: The described procedure lets this reviewer suspect that there might be a circular conclusion. Please provide evidence showing that this is not the case.

   Reply: We appreciate the insightful comment. As stated in the text (lines 322-323), the decision to use the threshold of $0.1$ g m$^{-3}$ was not arbitrary, but instead grounded on empirical observations from trial simulations. These simulations were benchmarked against independent LWC observations at the BNPP. We observed a consistent overestimation of the near-surface LWC by the WRF model when compared to these independent measurements. By implementing the stated threshold, our intention was to more closely align the model's outputs with observed data, in other words ensure that the model captures as much as the observed fog clouds as possible. We have now made this clear in the text (lines 328-332). It is important to note that while this threshold did influence certain aspects of our model's results, our conclusions were drawn from a broader analysis that is not solely reliant on this threshold. In other words, the threshold serves as a corrective measure to a known model bias but is not a foundational pillar upon which our conclusions rest.

7. Fig. 2: In the graphical representation, data of individual fog events overlap each other. Please separate individual events from each other.

   Reply: Thank you for your valuable feedback regarding the visibility of the lines in Figure 2. In our initial revision, a different reviewer highlighted the value of maintaining the line plots, emphasizing their relevance and utility for conveying specific aspects of our findings. As such, we have opted to retain Figure 2 in its current format to uphold the integrity of these insights. However, we deeply appreciate your concern regarding clarity and, to that end, a contour plot illustrating the diurnal variability in horizontal visibility, liquid water content, number concentration during the 12 fog cases is provided below for enhanced visualization and clarity. We hope this additional figure aids in the understanding of our analyses and we sincerely thank you for your astute observation.

[Figure]

(a) Horizontal visibility

(b) Liquid water content

(c) Number concentration

[Figure]

Figure R1. Diurnal variations in (a) horizontal visibility (km), (b) liquid water content (LWC; mg m$^{-3}$) and (c) number concentration of cloud droplets (cm$^{-3}$) for 12 fog events at BNPP

8. Fig. 5: Please be more specific on the shading. It is not clear what it exactly means.

Reply: Thank you for pointing it out. In Fig. 5b, the gray shaded area represents times when the visibility is less than 1 km (i.e. fog is present at the site). We have updated the caption of Fig. 5 in the revised manuscript to reflect this clarification.

9. Line 514 – 516: Agreed in principle. However, the reasoning is still speculative and must be classified as such in the manuscript.

Reply: We appreciate the reviewer's insight and agree with his/her comment. To address this, we have modified the section to better represent our findings and to reflect a more cautious interpretation in light of the available data (Line: 629:660). We now clarify the specific stage of the fog (peak of the mature stage) in which the observed phenomenon occurs. Also, added Figure 5c, Box plot showcasing the distribution of mass and number density for 12 fog events, both during the onset and mature stages of fog, distributed across four droplet-size ranges: 1-5 μm (black), 5-10 μm (red), 10-20 μm (blue), and 20-50 μm (green). Once again, we are grateful for the reviewer's constructive feedback and will make the necessary adjustments to the manuscript to ensure its clarity and scientific rigour.

10. Fig. 6a: The rH not even reaching 90 % in one of the events (no. 3?) needs explanation. Could it be a measuring artefact? If yes, what would that mean for the data of the other events?

Reply: Thank you for bringing this to our attention. Upon revisiting our data, it appears that the event the reviewer is referring to is event 4. For event 4, and despite its shorter duration of around 30 min (Table 3), the average RH during the fog event was still above 90%. As per reviewer suggestion, we have computed the mean RH and wind speed for all fog events and added them to Table 3. Your feedback is invaluable in ensuring the clarity and robustness of our findings. We are grateful for your meticulous review.

11. Fig. 6a: In view of this reviewer, it is not useful to compute and to show composite data as presented here, because the timing of the individual fog events differs largely.

Reply: Thank you for your feedback regarding the composite data presented in Fig. 6a. We understand your concerns about the differing timings of individual fog events potentially affecting the composite's accuracy and relevance. In light of your comment, we have removed the composite line from Figures 6a-b. We appreciate your input in helping improve the clarity and the rigor of our presentation.

12. Fig. 8: It is not clear what "turbulent vertical velocity" should be. It would be much netter to show the turbulent LWC fluxes or fog droplet number fluxes.

Reply: We apologize for the confusion regarding "turbulent vertical velocity" in Fig. 8. In this context, we used turbulent vertical velocity flux, to quantify the intensity of turbulence. However, upon reflection and in light of your feedback, we acknowledge that it would have been more appropriate to estimate the turbulence kinetic energy (TKE), which is a widely recognized metric for turbulence quantification. Consequently, we now present the TKE in Figure 8b. We would also like to point out that the LWC fluxes are already illustrated in Figures 9(a-c). Thank you for bringing this to our attention.

13. Fig. 9 shows that during a significant portion of the fog duration, the turbulent LWC fluxes are upward. Although this is briefly mentioned in a side comment in lines 620 and 621, and although this phenomenon has been observed by several authors at other locations, this phenomenon and the potential causes should be discussed in much more detail.

Reply: We would like to thank the reviewer for raising this issue. Indeed, several authors have reported upward turbulent LWC fluxes during fog, such as Degefie et al. (2015) for fog events around Paris during November 2012 - March 2013. It can occur in response to condensational growth near the ground surface and a subsequent broadening of the fog droplet size distribution. An inspection of Fig. 9 indicates this may also be the case in the

selected fog events. We have stated this in the text (lines 799-803) and would like to thank the reviewer again for his/her comment.

**References**

Bosveld, F. C. and Bouten, W.: Evaluation of transpiration models with observations over a Douglas-fir forest, Agricultural and Forest Meteorology, 108(4), 247–264, https://doi.org/10.1016/S0168-1923(01)00251-9, 2001.

Degefie, D. T., El-Madany, T.-S., Hejkal, J., Held, M., Dupont, J.-C., Haeffelin, M., and Klemm, O. (2015) Microphysics and energy and water fluxes of various fog types at SIRTA, France. Atmospheric Research, 151, 162-175. https://doi.org/10.1016/j.atmosres.2014.03.016.

Leclerc, M. Y. and Thurtell, G. W.: Footprint prediction of scalar fluxes using a Markovian analysis, Boundary Layer Meteorol., 52(3), 247–258, https://doi.org/10.1007/BF00122089, 1990.

Nelli, N. R., Temimi, M., Fonseca, R. M., Weston, M. J., Thota, M. S., Valappil, V. K., Branch, O., Wizemann, H.-D., Wulfmeyer, V. and Wehbe, Y.: Micrometeorological measurements in an arid environment: Diurnal characteristics and surface energy balance closure, Atmos. Res., 234, 104745, https://doi.org/10.1016/j.atmosres.2019.104745, 2020a.

Nelli, N., Francis, D., Fonseca, R., Bosc, E., Addad, Y., Temimi, M., Abida, R., Weston, M. and Cherif, C.: Characterization of the atmospheric circulation near the Empty Quarter Desert during major weather events, Front. Environ. Sci., 10, https://doi.org/10.3389/fenvs.2022.972380, 2022.

Stull, R. B., Ed.: An introduction to boundary layer meteorology, Springer Netherlands, Dordrecht., 1988.

---

## Author Comment (AC2)

**Response to Reviewer #2 comments on Manuscript egusphere-2023-956: Microphysics of radiation fog and estimation of fog deposition velocity for atmospheric dispersion applications**

**Major comments:**

In this paper, the authors have undertaken the task of characterizing radiation fog in around Barakah nuclear power plant in the United Arab Emirates. They have observed twelve fog events and conducted an analysis of microphysical and dynamical properties, and thermodynamics. Furthermore, they simulated a specific fog event (15 Feb. 2021) using various PBL schemes. The authors also delved into analyzing a particle dispersion model to comprehend the impact of fog deposition. Overall, this paper encompasses a wide range of analyses lack a thorough and cohesive understanding, making it challenging to arrive at a solid conclusion.

One notable concern is that fog events differ in each section and figure. For instance, section 3.1 focuses on the fog event of Feb 25, 2021 (Figure 5), while section 3.2 deals with fog events on Jan 27, Feb 4, and Feb 24, 2021 (Figures 6c and 6d). Similarly, section 4 concentrates on the Feb 16, 2021 fog event (Figure 7), whereas section 4 involves Jan 27, Feb 4, and Feb 24 (Figures 8 and 9). Section 5 centers around Feb 15. The reviewer is perplexed as to why different fog events are chosen for each section, leading to confusion regarding the author's primary focus. Furthermore, the connection between the WRF simulations and the preceding analyses (such as fog microphysical characteristics and the thermodynamic environment) is not well established. The instruments used to observe the fog events also appear to yield differing results.

In light of these issues, the reviewer strongly recommends that the authors narrow down their research topic and engage in more in-depth analyses. Addressing these concerns is crucial for enhancing the manuscript's coherence and overall quality. Given the current state of the manuscript, it is the opinion of the reviewer that it may not be suitable for publication in ACP.

Reply: Thank you sincerely for the detailed and insightful feedback. We recognize the significance of your concerns about the coherence and specificity of the analyses of various fog events, as well as the WRF simulations. Your suggestions for narrowing and deepening our research focus have been meticulously considered and addressed in the revised manuscript by considering all the 12 events throughout the manuscript except for the modelling part which is done for only 4 events due to computational requirement to run for the 12 cases. More specifically:

- In Section 3.1, we have updated our discussion, presenting findings from all observed fog events and articulating the common features and unique distinctions among them. Comprehensive insights from these events are now encapsulated in the updated Figs. 2-4 and 5c;

- In Section 3.2, the diurnal variations in relative humidity and wind speed have been scrutinized for all 12 fog events, as visualized in the revised Figs. 6a-b. We have enriched our discussion of the role of temperature inversions on Liquid Water Content (LWC) and number concentrations, providing detailed insights through representative temperature and specific humidity profiles in Figs. 6c-d. Additionally, the inversion strength analyses for all fog events is now depicted in Fig. A2;

- Regarding Section 4, due to the unavailability of eddy covariance data for the fog events in 2021, only fog events occurring in 2022 are used to derive fog deposition velocity, Turbulent Kinetic Energy (TKE), and Liquid Water Flux. This is explicitly mentioned in the text (Line: 736-740).

In connecting the analysis of fog microphysics with WRF simulations, we need specific details from fog microphysics, like the mean volume diameter and liquid water content, to estimate the fog deposition velocity. Through the analysis of microphysics, we first determine the fog deposition velocities. These determined values are then integrated into FLEXPART to assess the deposition of radioactive materials. In the revised version of our manuscript, we have incorporated the WRF simulation results for fog cases from 2022. For these 2022 fog events, we calculated the average fog deposition velocity for each instance and subsequently quantified the radionuclides deposition rates, which are illustrated in Figure 12. For instance, the average deposition velocity on January 27, 2022, was 7.87 cm s$^{-1}$. On February 4, 2022, it registered at 3.51 cm s$^{-1}$, and on February 24, 2022, it was 2.11 cm s$^{-1}$. These velocities were employed to ascertain the deposition rate of radionuclides. We have included this information in the revised manuscript (End of section 4).

We do believe these amendments and clarifications enhance the manuscript and address the pivotal points you have raised. Your insights have been instrumental in refining the quality and rigor of our study.

**Minor comments:**

1. Line 76: Remove the semicolon (;)

Reply: corrected (Line: 81)

2. Line 90-91: I am not clear what the sentence means. Also, it might need to consider a referent to support author's argument.

Reply: In the revised version of the manuscript we have rephrased the referred sentence accordingly (lines 89-91). Rephrased sentence is given below,

"Although previous analyses of fog in the UAE have been conducted using satellite data (Weston and Temimi, 2020a) and in-situ measurements for a single fog event (Weston et al., 2022), a comprehensive analysis of fog microphysics and dynamics has yet to be carried out in this region."

3. Line 101: If the author would like to emphasize this (first observation-derived fog deposition velocity for the UAE), it would be great if the author presents a distinct feature about the first study by comparing the previous studies with different regions.

Reply: Thank you for your insightful suggestion regarding emphasizing the distinctiveness of our study – the first to derive fog deposition velocity based on observations in the UAE. In comparison to previous studies in different regions, our observed fog deposition velocities range from 2.1 to 7.8 cm s$^{-1}$ (Figure 8a). These findings are comparable in order of magnitude with the values reported by Tav et al. (2018), particularly for the bare soil category. However, they are notably smaller than the 16 cm s$^{-1}$ and 40 cm s$^{-1}$ observed for cypress and grass surfaces respectively in the aforementioned study. This comparative analysis underscores the variation in fog deposition velocities across different geographical regions and surface types, highlighting the unique contribution and criticality of our study in understanding these dynamics within the UAE.

4. Figure 1: It could be very helpful the author can mark the observational sites on the map, because the instruments installed the different sites. Also, please change the color for the ocean, because it looks land since the lowest elevation over land also shaded blue color.

Reply: We genuinely appreciate your meticulous observation concerning Figure 1 and wholeheartedly agree that marking the observational sites and adjusting the color scheme will enhance the clarity and in formativeness of the figure. In the revised Figure 1, we have now annotated the observational sites clearly, ensuring that readers can effortlessly identify the locations where instruments were installed. Furthermore, we have modified the color scheme to eliminate any confusion between the ocean and land areas with low elevation, by opting for a distinct color for the ocean, ensuring an unequivocal differentiation between the two. These enhancements aim to facilitate a more intuitive and insightful interpretation of the geographic contexts relevant to our study. We trust that these adjustments will amplify the figure's utility and clarity for our readers and sincerely thank you for pointing out these opportunities for improvement.

5. Table 1: It could be very helpful if the author can add the measurement bias at each instrument.

Reply: We greatly value your suggestion regarding the enhancement of Table 1 by incorporating the measurement accuracy of each instrument. Acknowledging the significance of transparency and thoroughness in presenting instrument data, in the revised Table 1, we have provided the measurement accuracy for each respective instrument. Your pointed suggestion substantially contributes to the improvement of our manuscript by enhancing the depth and reliability of the information presented. Thank you once again for your prudent observation.

6. Table 1: There were two analysis period in one box. Please modify the box correctly. The period was not same among the instruments. How the author analyzes the data? Does them use the same period of case?

Reply: Thank you for your insightful comment regarding the analysis periods outlined in Table 1. We acknowledge the variability in the measurement periods for the instruments utilized, largely due to the challenges posed by conducting field measurements in a hyper-arid environment. In this study, our analysis strictly employs data from the 12 fog events detailed in Table 3. Additionally, we sought not only to judiciously utilize available data but also to highlight the data availability periods for the international research community. This approach aims to promote transparency and foster potential collaborative insights in future research endeavors. We trust that this clarification and additional context regarding our methodology and data usage will provide a clearer understanding of our approach, alleviating any potential concerns. We extend our appreciation for your diligent scrutiny, which undoubtedly enhances the transparency and comprehensiveness of our manuscript.

7. Line 221: Why the author uses the WSM3 rather than WSM6 as microphysics scheme? The WSM6 provides the 6 class hydrometeor types including ice, snow, and graupel.

Reply: We would like to thank the reviewer for raising this issue. The physics schemes selected in this work are those found to be optimal for western UAE following the sensitivity experiments conducted by Abida et al. (2022). We have clarified this in the text (Lines: 257-260, section 2.4.1).

8. Line 289-290: Form what?

Reply: Thank you for pointing out the concern on Line 289-290. We infer that you are seeking a reference for the fog definition provided. In response, we have cited the relevant source, the World Meteorological Organization report (WMO, 2008), in the revised manuscript to validate the definition utilized. We appreciate your suggestion, which enhances the accuracy and credibility of our work.

9. Figure 2: at Barakkah? Does this mean at BNPP?

Reply: Thank you for catching that oversight. We apologize for any confusion caused by the use of "Barakah." You are correct, it does indeed refer to BNPP. To prevent any future misinterpretations, we have clarified this by explicitly mentioning BNPP in the captions of Figure 1 and Figure 2 in the revised manuscript. We appreciate your diligence and your constructive suggestion.

10. Line 509-511: Need result/figure or reference to support the argument.

Reply: We have added two references Tav et al. (2018) and Weston et al. (2022 to back up our statement. (Lines: 641-674).

11. Figure 6a,b: Hard to see the lines

Reply: Thank you for your valuable feedback regarding the visibility of the lines in Figure 6a,b. In our initial revision, a different reviewer highlighted the value of maintaining the line plots, emphasizing their relevance and utility for conveying specific aspects of our findings. As such, we have opted to retain Figure 6ab in its current format to uphold the integrity of these insights.

However, we deeply appreciate your concern regarding clarity and, to that end, a contour plot illustrating the diurnal variability in Relative Humidity (RH) and wind speed during the 12 fog cases is provided below for enhanced visualization and clarity. We hope this additional figure aids in the understanding of our analyses and we sincerely thank you for your astute observation.

[Figure]

(a) Relative Humidity

(b) Wind speed

[Figure]

Figure R1.  (a-b) Diurnal variations (18 LT - 11 LT) in relative humidity (RH; %) and wind speed (m s$^{-1}$) for all fog events.

12. Line 623-624: How can see this result? The reviewer can not find the result correctly in Figure 9a. The background map showed wide ranges. Please explain more about this.

Reply: We thank the reviewer for raising this issue and apologize for the poor wording of the text. The largest contribution to the total LWC (black line) comes from the droplets of sizes 20-45 μm (darkest blue shading in the background), cloud droplets of smaller sizes play a much reduced role as evidenced by the lighter shading. We have rephrased the referred sentence for clarity in the revised version of the manuscript (Lines: 803-805).

13. Line 677-678: What the author refers to? FLEXPART?

Reply: No, we are referring to the convention used in Fig. 9 regarding the total liquid water flux. We have rephrased the text for clarity (Lines: 793-794).

14. Figure 10: Why the author shows only two simulations (ACM2 and MYJ)? The simulations were performed five (Table 2), so please add more figures for all simulations.

Reply: Following the reviewer's suggestion we now show the results for all five PBL schemes considered in Fig. 10 and have expanded the discussion in the text (Section 5).

**References**

Abida, R., Addad, Y., Francis, D., Temimi, M., Nelli, N., Fonseca, R., Nesterov, O. and Bosc, E.: Evaluation of the Performance of the WRF Model in a Hyper-Arid Environment: A Sensitivity Study, Atmosphere, 13(6), 985, https://doi.org/10.3390/atmos13060985, 2022.

Tav, J., Masson, O., Burnet, F., Paulat, P., Bourrianne, T., Conil, S. and Pourcelot, L.: Determination of fog-droplet deposition velocity from a simple weighing method, Aerosol and Air Quality Research, 18, 103–113, 2018.

WMO, 2008: Aerodrome reports and forecasts: A user's handbook to the codes. World Meteorological Organization, 81 pp.

Weston, M., Francis, D., Nelli, N., Fonseca, R., Temimi, M. and Addad, Y.: The first characterization of fog microphysics in the united arab emirates, an arid region on the arabian peninsula, Earth and Space Science, 9(2), https://doi.org/10.1029/2021EA002032, 2022.

---

## Author Comment (AC3)

**Response to reviewer #3 comments on 'Microphysics of radiation fog and estimation of fog deposition velocity for atmospheric dispersion applications' By Abida et al., 2023**

The authors would like to thank the editor and the reviewers for their comments which helped to improve the manuscript. Kindly find below our response point-by-point in blue.

**General**

In this paper the authors investigate a series of fog events near the nuclear power station Barakah, situated close to the Arabian Sea in an arid environment. The investigation is performed using quite a number of different meteorological instruments on-site and numerical simulations, employing and testing various parameterizations for the planetary boundary layer physics. The final goal is to show whether and how much fog would change the contamination of soil nearby if there would be a nuclear accident with release of radioactive matter. Indeed it is found that fog changes the deposition pattern of the radioactive material. It is said that the results depend on the type of the ground and that therefore the present results are peculiar for the considered site. It is recommended that fog deposition is considered as an additional scavenging mechanism in dispersion models.

The paper is well written and the reader has the nice feeling of a fluent read. My impression is that there are no technical flaws or unjustified statements. Most statements are put into context by mentioning results from similar studies. To my view, this paper can be published after some minor corrections which I think would improve an already quite good manuscript.

We thank the reviewer for taking the time to read our manuscript and provide invaluable feedback.

**These are my minor comments:**

1. Line 47 ff: Please rewrite "Masson et al. (2015) showed that cloud water was relevant to detect 134Cs (.....) on a longer time scale than both in aerosol and in rainwater." I don't understand this sentence. Additionally, check whether it is 134Cs or rather 137Cs.

   REPLY: Thank you for your feedback. We agree that the sentence was not clear and have rewritten it for clarity (Lines: 51-53). Also, we confirm that we intended to reference $^{134}$Cs, which is known to have a longer detection time scale in cloud water. The revised sentence is as follows: 'Subject to the capabilities of trace level measurement, Masson et al. (2015)

found that the detection of $^{134}$Cs (a radionuclide released during accident with a half-life of 2.06 year) was possible over a longer time scale in cloud water, compared to its detection in aerosols and rainwater.

2. Line 55: "the number of studies on fog in arid and semi-arid regions has caught recent attention". Please rewrite, as it says that the number of something has caught attention which is surely not meant.

   REPLY: Thank you for pointing out the ambiguity in the original phrasing. We have revised the referred text to clarify that the recent attention has been focused on the topic of fog in arid and semi-arid regions, rather than the number of studies themselves. The revised sentence now reads, 'While there has been growing interest in the study of fog in arid and semi-arid regions (Eckardt and Schemenauer, 1998; Feigenwinter et al., 2020; Katata et al., 2010; Spirig et al., 2021), research regarding fog deposition of radionuclides in such environments is notably lacking.' (Lines: 59 - 62)

3. Line 137: I am not happy with the expression "to measure fog microphysics". Perhaps better "to observe and quantify microphysical processes".

   REPLY: Corrected (Lines: 147-148)

4. Line 146: Has MVD already been spelled out?

   REPLY: Thank you for pointing out the usage of the acronym 'MVD'. We realize that it had not been spelled out before line 146. In the revised manuscript, we have introduced the term 'Median Volume Diameter (MVD)' at its first occurrence to ensure clarity for the readers (Line: 159).

5. Line 147: "2 counts cm-3": Unclear. Either it must be 2 counts per ccm and unit time, or you must add over which time the counting had been integrated.

   REPLY: Thank you for your comment. We understand the confusion regarding the unit of measure. We have clarified this in the revised manuscript to indicate that the '2 counts cm-3' is an instantaneous count per second. The revised sentence now reads as follows: '...we only considered values where the number concentration was greater than 2 counts cm-3 sec-1.'. (Lines: 160).

6. Line 156/7: Incomplete sentence "For a full description of the operational method and measurement uncertainties (..)."

REPLY: We appreciate your observation regarding the incomplete sentence. We have now revised the sentence in the revised version of the manuscript (Lines: 169-170)

7. Line 172: Please find a better expression than "construction structures".

REPLY: Corrected (Line: 183-184)

8. Line 214: Correct "Forty_-five".

REPLY: Corrected (Line: 253)

9. Line 223: Please rewrite "the cloud water-rain and the cloud ice-snow are treated for temperatures above and below 0°C."

REPLY: Thank you for your feedback. In line with your suggestion, we have revised the sentence in line 223 to more clearly distinguish the treatment of liquid and frozen water hydrometeors at different temperatures (Lines: 262-265).

10. Line 399: Not sure whether METAR was defined somewhere.

REPLY: Thank you for pointing out the usage of the acronym 'METAR'. We realize that it had not been spelled out before line 399. In the revised manuscript, We have introduced the term 'Meteorological Aerodrome Report (METAR)' at its first occurrence to ensure clarity for the readers (Lines: 529-530).

11. Line 403/4: "it is observed that similar to the horizontal visibility, the LWC and number concentration also increase abruptly". This should be rewritten since it indicates that visibility would increase which it doesn't.

REPLY: Thank you for pointing out the potential for misunderstanding in our original phrasing. We now revised line sentence to clarify that as visibility decreases, the LWC and number concentration increase (Lines: 533-535).

12. Figure 2 should be replaced. The presentation of the data is misleading. The problem is that the data from independent days are shown in the form of a contour plot (or something similar) which results in smooth transitions between these days. Instead, we have 12 independent time series and I strongly suggest replacing the plot with one showing simply the 12 timeseries as 12 single curves.

REPLY: We thank the reviewer for pointing out this. We understand your concern about the potential for misrepresentation when using the contour plot to represent the independent time series data. As you suggested we now revised the Figure 2 to clearly depict the 12 independent time series as individual curves.

13. Line 416 and several other instances: You write "bin size" but you mean "size bin" which are two different things. Additionally, I do not remember whether you defined before what "size" is. Is it the droplet diameter or radius? Please check.

REPLY: Thank you for pointing out the need for clarity in our terminology. We have taken your suggestion and will use 'size bin' when referring to categories of droplet diameters. For clarity, we've added a definition early in the manuscript specifying that 'size' refers to the droplet diameter in our study (Line: 155-156). We appreciate your attention to these details.

14. Line 429: Delete "While" and start the sentence "The second mode...".

REPLY: Corrected (Line: 561)

15. Lines 432/3: Please check the numbers 4.5 vs 5.5 micron. As I understood, there is only one small mode that should be characterised by only one value.

REPLY: Thank you for drawing out attention to this discrepancy. It was a typographical error. The correct value should be 4.5 micron, not 5.5. Corrected in the revised version of the manuscript (Line: 563-566).

16. Line 437: I suggest replacing "discrepancies" with "differences". Discrepancies implies to me that there is something inconsistent, but that seems not implied here.

REPLY: Corrected (Line: 570)

17. Line 474: It seems that onset/termination times and durations are inconsistent here and in table 3.

REPLY: Thank you for bringing this inconsistency to our attention. The fog case on 16th Feb included two distinct fog events, which should have been represented separately in Table 3. We now have updated Table 3 in the revised manuscript. Same is corrected in the text (Lines: 605-608).

18. Line 477: "droplet sizes between 20 and 30". Add units.

    REPLY: Units added (Line: 610)

19. Figure 3: x-axis label should be radius or diameter, but not bin.

    REPLY: Corrected (Figure 3).

20. Line 528: Check "number concentration were observed Figs. 2a-c". I suggest also that you would better write "number concentrations were measured".

    REPLY: Thank you for your suggestion. We agree that using 'measured' is more appropriate and precise in this context. We now corrected this in the revised version of the manuscript (Line: 687).

21. Line 534 and Figure A2: Use "backscatter profiles" instead of "backscattered profiles" (The profiles themselves are not backscattered).

    REPLY: Thank you for your suggestion regarding the use of terminology. We agree that 'backscatter profiles' is a more accurate description. We now corrected in the text (Line: 707-708) and Figure A2 caption accordingly to replace 'backscattered profiles' with 'backscatter profiles'. We appreciate your attention to detail.

22. Lines 579/80: Please rewrite "Settling velocities calculated every 5 min and values with visibility greater than 1 km are discarded (Fig. 7b)." I don't understand what you mean.

    REPLY: Thank you for your comment. We understand the confusion caused by our original sentence. We have revised it for clarity (Lines:753-754) and it now reads: 'We calculated settling velocities every 5 minutes, discarding the values associated with visibility greater than 1 km (Fig. 7b).' We hope this addresses your concern.

23. Line 588: What is an "increasing relationship"?

    REPLY: Apologies for any confusion caused by our use of the term 'increasing relationship.' We intended to refer to a direct relationship, where an increase in Liquid Water Content (LWC) would correspond with an increase in the gravitational settling rate. We have revised the sentence to read: 'This calculation confirms that a direct relationship between LWC in the fog and the gravitational settling rate is not necessarily warranted.' (Line: 761-765). We appreciate your feedback for improving the clarity of our manuscript.

24. Line 681: Here you should repeat or mention the PBL schemes that you use, for the convenience of the reader. A few words on each PBL schemes particular properties and abilities would be helpful as well.

REPLY: We thank you for your insightful remark. Indeed, a reminder of the PBL schemes used is necessary here, and accordingly the text is revised as follows:'Fog simulation is sensitive to the choice of the PBL parameterization scheme. Therefore, we performed five 48-hour WRF simulations, using three local PBL schemes, MYJ, MYNN2.5 and MYNN3.0, and two non-local PBL schemes, YSU and ACM2.' (Line: 891-894, section 5). The properties and abilities of each scheme are now added to the Table 2. Also some discussion about their characteristics can be found in lines 281-289'.

25. Line 697: Replace "ribbon" with "band".

REPLY: We appreciate your suggestion to replace 'ribbon' with 'band'. We agree that 'band' is a more appropriate term in this context. We have made this correction in the revised manuscript (Line: 923-924, section-5).

26. Line 707: Please use a simple dot or an "x" instead of a star for multiplication. In the figures it is a "x" which is ok.

REPLY: Corrected.

27. Line 734: write "long" instead of "longer".

REPLY: Corrected.

28. Lines 773 ff: Please rewrite "the actual impact of fog on radionuclide deposition can vary widely depending on the specific situation as well as more on the solubility and chemical form of the radionuclide-labeled particles." I don't understand the second half of the sentence beginning from "as well as...".

REPLY: Thank you for your feedback. We realize our sentence was unclear. We have rewritten it to better communicate our point: 'The actual impact of fog on radionuclide deposition can vary widely depending on the specific situation. Moreover, the solubility and chemical form of the radionuclide-labeled particles can significantly influence this impact.'

29. Line 807: What is a "size distribution of the mean number of droplets"?

REPLY: Thank you for your feedback. We understand the original phrase was unclear. To enhance the clarity, we have changed it to 'size distribution of mean number concentration' for each fog event.'.